# The GRISLI-LSCE contribution to ISMIP6, Part 1: projections of the Greenland ice sheet evolution by the end of the 21[st] century

Aurélien Quiquet[1] and Christophe Dumas[1]

[1]Laboratoire des Sciences du Climat et de l'Environnement (LSCE), UMR8212, CEA/CNRS-INSU/UVSQ, Gif-sur-Yvette Cedex, France

*Correspondence to:* A. Quiquet (aurelien.quiquet@lsce.ipsl.fr)

**Abstract.**

Polar amplification will result in amplified temperature changes in the Arctic with respect to the rest of the globe making the Greenland ice sheet particularly vulnerable to global warming. While the ice sheet has been showing an increased mass loss in the past decades, its contribution to global sea level rise in the future is of primary importance since it is at present the largest single source contribution after the thermosteric contribution. The question of the fate of the Greenland and Antarctic ice sheets for the next century has recently gathered various ice sheet models in a common framework within the Ice Sheet Model Intercomparison Project for CMIP6. While in a companion paper we present the GRISLI-LSCE contribution to ISMIP6-Antarctica, we present here the GRISLI-LSCE contribution to ISMIP6-Greenland. We show an important spread in the simulated Greenland ice loss in the future depending on the climate forcing used. The contribution of the ice sheet to global sea level rise in 2100 can be thus as low as 20 mm of sea level equivalent (SLE) to as high as 160 mm SLE. Amongst the models tested in ISMIP6, the CMIP6 models produce larger ice sheet retreat than their CMIP5 counterparts. Low emission scenarios in the future drastically reduce the ice mass loss. The oceanic forcing contributes to about 10 mm SLE in 2100 in our simulations. In addition, the dynamical contribution to ice thickness change is small compared to the impact of surface mass balance. This suggests that mass loss is mostly driven by atmospheric warming and associated ablation at the ice sheet margin. With additional sensitivity experiments we also show that the spread in mass loss is only weakly affected by the choice of the ice sheet model mechanical parameters.

## 1 Introduction

The relative contribution of land ice to global mean sea level rise has considerably increased in the recent decades, and is now larger than the thermosteric effect (Nerem et al., 2018). Amongst the different contributions, the Greenland and Antarctic ice sheets have a potential to raise substantially the global mean sea level, with a weakly constrained trajectory (Oppenheimer et al., 2019). While observational datasets show a dramatic increase in mass loss over the last decades for both ice sheets (Mouginot et al., 2019; Rignot et al., 2019), there is an urgent need for robust assessment of future sea level rise by projections obtained with numerical models.

Most of the time, these projections involve comprehensive ice sheet models that compute the ice thickness change that results from evolving forcings, such as climate change. On top of uncertainties related to future climate evolution, there are important differences amongst existing ice sheet models, and these differences represent a major source of uncertainty for the fate of the ice sheets in the future. First, in order to save computing time, most of the ice sheet models use various asymptotic expansions (e.g. the shallow ice and shallow shelf approximations or higher-order models) even if, more recently, models that account explicitly for all the stress components of the Stokes equation at the ice sheet scale have emerged (e.g. Seddik et al., 2012). This difference in terms of ice sheet model complexity is a source of uncertainty for future projections. Second, ice sheets respond to a wide spectrum of timescales, from sub-annual to multi-millenial. As a result, diverse methodologies to initialise the models for projection purposes have been developed. For Greenland ice sheet models, these differences in methodologies lead to an even larger uncertainty for future projections than model complexity and explain most of the multi-model spread (Goelzer et al., 2018). A last source of uncertainty lies in poorly known processes, such as sub-glacial processes, or processes that are not included in models due to their complexity or too fine spatial scale, such as outlet glacier dynamics or fracturing. Large international intercomparison exercises are a useful way to quantify these different uncertainties and to infer robust sea level projections into the future.

The Ice Sheet Model Intercomparison Project for CMIP6 (ISMIP6, Nowicki et al., 2016), endorsed by the Coupled Model Intercomparison Project – phase 6 (CMIP6), aims at investigating the role of dynamic Greenland and Antarctic ice sheets in the climate system and to reduce the uncertainty in ice sheet contribution to global sea level rise in the future. Within this framework, stand-alone ice sheet model experiments have recently been carried out by world-wide research groups. Many model experiments using both CMIP5 and CMIP6 climate forcing scenarios until 2100 were conducted with ice sheet models spanning a range of model complexities and using different initialisation techniques. To date, this is the most ambitious intercomparison exercise dedicated to the fate of the Greenland and Antarctic ice sheets in the future. At the Laboratoire des Sciences du Climat et de l'Environnement (LSCE), we participated to this stand-alone intercomparison with the GRISLI model (Quiquet et al., 2018). This model uses the shallow ice and shallow shelf approximations and is relatively inexpensive in terms of computational cost. We were thus able to perform all the different experiments of ISMIP6.

The aim of this paper is to discuss the role of the forcing uncertainties for future projections of the Greenland ice sheet contribution to global sea level rise when using our model. This individual model response can be put in perspective with respect to the multi-model spread discussed in Goelzer et al. (2020). This paper discusses additional experiments not included in the community paper (CMIP6 forcing, separate effects of the oceanic and atmospheric forcings). Compared to Goelzer et al. (2020), we provide here a more detailed description of the initial state and its associated biases and model drift. A companion paper (Quiquet and Dumas, 2020a) describes the results for the Antarctic ice sheet.

In Sec. 2 we describe briefly the GRISLI ice sheet model as well as the procedure used for its initialisation. We also provide information on the ISMIP6-Greenland forcing methodology and we provide an overview of the different experiments performed.

In Sec. 3 we discuss the results for the different experiments in terms of geometry and dynamical changes. We discuss these results in a broader context in Sec. 4 and we conclude in Sec. 5.

## 2 Methods

### 2.1 Model and initialisation

For this work, we use the GRISLI ice sheet model. The model is a 3D thermo-mechanically coupled ice sheet model that solve the mass conservation and force balance equations. The model is fully described in Quiquet et al. (2018) and we only provide here a brief overview of its characteristics.

Assuming incompressibility, the mass conservation equation for a grid element is:

$$\frac{\partial H}{\partial t} = M - \nabla \left( \bar{\mathbf{U}} H \right), \tag{1}$$

with $H$ the local ice thickness, $M$ the total mass balance and $\bar{\mathbf{U}}$ the vertically averaged horizontal velocity vector. $\nabla \left( \bar{\mathbf{U}} H \right)$ is thus the ice flux divergence.

The Stokes momentum equation is solved using asymptotic zero-order expansions. For the whole geographical domain, we assume that the total velocity is the sum of the velocities predicted by the two main approximations: the shallow ice approximation (SIA) in which the deformation is entirely driven by the vertical shear and the shallow shelf approximation (SSA) in which the vertical shear is neglected and the horizontal stresses are predominant. Practically, this means that we use the SSA equation as a sliding law (Bueler and Brown, 2009; Winkelmann et al., 2011). Grounded cold base and floating shelves are special cases for which there is infinite friction at the base or none, respectively. Elsewhere, friction is assumed to follow a Weertman (1957) power law with a till layer that allows viscous deformation:

$$\tau_{\mathbf{b}} = -\beta \, \mathbf{U_b}, \tag{2}$$

where $\tau_b$ is the basal drag, $\beta$ is the basal drag coefficient and $\mathbf{U_b}$ is the basal velocity. The basal drag coefficient is spatially variable but constant in time (except in specific cases such as during the inversion procedure).

Like most ice sheet models, GRISLI uses a flow enhancement factor that increases the ice fluidity in the SIA (Quiquet et al., 2018). However, here we use a flow enhancement factor set to 1 (no enhancement). Similarly, the flow enhancement factor for the SSA is also set to 1.

Similarly to what has been done with GRISLI for the initMIP-Greenland experiments (Goelzer et al., 2018), we used here an inverse procedure to initialise the model at the start of the historical experiment. We mostly followed the iterative method of Le clec'h et al. (2019b) which consists at yielding the map of the basal drag coefficient $\beta$ that minimises the ice thickness error with respect to observations. To this aim, we first run a 30 kyr experiment with fixed topography and perpetual present-day climate forcing in order to compute the thermal state of the ice sheet in agreement with the boundary conditions. From this, we do multiple 200-yr long experiments under constant present-day climate forcing but with an evolving topography. During the

first 20 years of these experiments, we adjust the basal drag coefficient to minimise the ice thickness mismatch with respect to the observations. Each 200-yr iteration uses exactly the same initial condition for the ice thickness and temperature but have a different initial basal drag coefficient. Also, the ice thickness error at the end of the 200-yr long experiment is used to facilitate convergence towards the observed ice thickness through a local basal drag modification. This basal drag coefficient modifica-

tion consists of finding an ice flux on the simulated topography as close as possible to the balance ice flux on the observed topography. After a few 200-yr experiments, we repeat the thermal equilibrium computation restarting from the previous equilibrium state with the newly inferred basal drag coefficient. In doing so, the basal drag coefficient and the temperature at the base are consistent with each other. For this work we performed more than ten thermal equilibrium experiments, each one followed by five iterations of 200 years.

At the end of the iterative process, we use the last inferred basal drag coefficient together with the corresponding thermal state to run a short relaxation experiment of 20 years. The end of this relaxation experiment defines our initial state which is used to begin the historical experiment *hist* and the control experiment *ctrl* (see Sec. 2.3).

Our ice thickness and bedrock topography of reference is the BedMachine v.3 (Morlighem et al., 2017). This dataset is used as a target for our iterative procedure to infer the basal drag coefficient. It is also used as the starting topography for the short relaxation that defines our initial state. Our present-day reference climate forcing, namely annual near-surface air temperature and annual surface mass balance, comes from the MAR v3.9 (Fettweis et al., 2013, 2017) forced at its boundary by MIROC5, averaged over the 1994-2015 period. On top of this climate forcing, we also add a strongly negative surface mass balance term

of -15 m yr$^{-1}$ outside the present-day ice mask in the observational dataset in order to avoid inconsistencies between the climate forcing and the initial ice sheet geometry. This present-day reference climate forcing is used for the initialisation procedure and for the control experiment *ctrl*. The model is run on a Cartesian grid at 5 km resolution covering the Greenland ice sheet using a stereographic projection. Since 5 km is too coarse to represent Greenland's floating ice tongues, the sub-shelf melting rate has been set to a large value (200 m yr$^{-1}$) to discard simulated floating points. Glacial isostatic adjustment has been deactivated

for all the experiments shown in this manuscript.

## 2.2 ISMIP6-Greenland forcing methodology

The ISMIP6-Greenland working group distributed atmospheric and oceanic forcings to drive individual ice sheet models. They also suggest a forcing methodology so that participating models are run using a common framework. Full description of the methodology is available in Nowicki et al. (2020) and only a summary is presented here.


For the atmospheric forcing, MAR v3.9 has been run from 1950 until 2100 forced at its boundaries by a selection of CMIP5 and CMIP6 general circulation model (GCM) outputs. To force the ice sheet models, yearly anomalies of near-surface air temperature and surface mass balance are provided. These anomalies were constructed as the difference of a given yearly value with the climatology over the reference period 1960-1989. In addition, to account for the surface elevation feedback on temperature

and surface mass balance, yearly values of vertical gradients for these two surface variables are also provided. These spatially variable gradients were evaluated with the MAR model with the method of Franco et al. (2012).

Ice–ocean interactions for the Greenland ice sheet are most of the time poorly represented amongst ISMIP6-Greenland participating models. This is mostly due to the fact that the spatial scale needed to represent such interactions is out of reach for most models. This is also the case for GRISLI, where the 5 km resolution grid is too coarse to capture marine-terminating outlet glaciers. To cope with this problem, retreat masks for outlet glaciers have been made available in ISMIP6-Greenland. They were obtained with simple parametrisations calibrated and tested against observational datasets (Slater et al., 2019). These masks provide, for a given resolution, the fraction of the grid that becomes ice free and they are used to impose a specific retreat rate of the marine front. For each climate forcing, three retreat masks are available for different oceanic sensitivities (*low*, *medium* and *high*). Since our model does not account for partially glaciated grid cell, the fractional information given by the retreat masks is used to reduce the local ice thickness with respect to a reference ice thickness (i.e. the ice thickness evolution for the outlet glaciers is imposed). The reference ice thickness could have been chosen as the ice thickness at a specific time (e.g. the ice thickness at the end of the historical experiment). However, in doing so, we may create strong discontinuities in ice thickness when the retreat mask is used for the first time. For this reason, we choose instead the value of the local ice thickness at the time when the imposed retreat starts to play as a reference ice thickness.

## 2.3 List of experiments

The ice sheet state inferred at the end of the initialisation procedure (Sec. 2.1) is used as initial condition for the historical experiment *hist*. In our case, the historical experiment starts in January 1995 and ends in December 2014. For this historical experiment, we use the climate forcing of MAR forced at its boundary by the MIROC5 climate model. The projection experiments described in the following are all branched from the end of the year 2014 of this historical experiment and span 2015-2100 (86 simulated years).

ISMIP6-Greenland listed a large ensemble of experiments to be performed with individual ice sheet models (Tab. 1). The ensemble of experiments is large enough to assess: ice sheet sensitivity to the chosen climate forcing, CMIP5 with respect to CMIP6, sensitivity to the greenhouse gas emissions scenarios, the respective role of oceanic forcing with respect to atmospheric forcing, and to quantify the uncertainty regarding the oceanic forcing. The core experiments (Tier 1) consist of a selection of three CMIP5 climate models (MIROC5, NorESM and HadGEM2-ES) run under the RCP8.5 scenario for greenhouse gases. In addition, MIROC5 was chosen to be run with a different RCP scenario (RCP2.6) and using different oceanic sensitivities (*high* and *low* in addition to *medium*). Tier 2 has two subsets: an extended ensemble with three additional CMIP5 models using RCP8.5 and an other with four CMIP6 models. Amongst CMIP6 models, CNRM-CM6 has been run under two scenarios: a high (SSP585) and a low (SSP126) emission scenario. Tier 3 has also two subsets. The first one aims at quantifying the respective role of the oceanic forcing with respect to atmospheric forcing, running the ice sheet models with only one of this forcing

at a time. Three climate models were selected (MIROC5, CSIRO-Mk3.6 and NorESM) and as in Tier 1, MIROC5 was run for two greenhouse gases scenarios and different oceanic sensitivities. Finally, the second subset of Tier 3 contains the ten climate models (CMIP5 and CMIP6) each time run with the two additional oceanic sensitivities (*high* and *low*). CNRM-CM6 is the only one in this subset that has run under two emission scenarios (SSP585 and SSP126). We performed all these experiments with the GRISLI ice sheet model.

In addition to these projection experiments, we also perform two control experiments in which the climate forcing remains unchanged, being our reference climate forcing used during the initialisation procedure (zero anomaly). The control experiment *ctrl* starts from the initial state resulting from our initialisation procedure and covers the 1995-2100 period (106 years). The *ctrl_proj* experiment starts in January 2015, like the projection experiments, and runs for 86 years under a constant climate forcing. The *ctrl* experiment can be used to quantify the simulated model drift over the whole time period (1995-2100). By contrast, the *ctrl_proj* can be directly used to quantify the importance of climate forcing evolution since it uses the same initial state in 2015 as the different projection experiments.

## 3   Results

We aim here at providing a detailed description of the historical experiment *hist* and the model response under the various forcings of the projection experiments. While some information is given in this section, the reader is invited to refer to Goelzer et al. (2020) to compare in details the response of GRISLI to other participating models. A map of Greenland with the names of the major ice streams discussed in the following is shown in Fig. 1.

### 3.1   Present-day simulated ice sheet

At the end of the historical experiment *hist*, with a value smaller than 30 m, GRISLI shows the lowest ice thickness root mean squared error (RMSE) with respect to the observations of Morlighem et al. (2017) amongst the ISMIP6-Greenland participating models (Goelzer et al., 2020). This is a result of the initialisation procedure we use (Sec. 2.1) that includes only a short relaxation of 20 years. With an historical experiment of 20 years only, the model has no time to depart strongly from the observations. The map of the ice thickness difference with respect to observations is shown in Fig. 2a. The model shows a very good agreement with the observations for most of the ice sheet, except at specific locations at the margin. In particular, South-East Greenland is the least well reproduced with local errors greater than 200 metres. In the region of Kangerdlugssuaq and Helheim glaciers, there is an ice thickness overestimation near the glacier termini and an underestimation upstream. These differences with the observations can be due to the fact that this area is particularly difficult to model since it has a complex surface mass balance pattern with very strong horizontal gradients and also a rough topography that is not necessary well captured at 5 km resolution.

Some of the ice thickness mismatch with respect to the observations can be partly explained by error related to ice dynamics. GRISLI has indeed an ice velocity RMSE with respect to the observations (Joughin et al., 2016) of about 35 m yr$^{-1}$, making the model the sixth worst model out of 21 (Goelzer et al., 2020). Our initialisation procedure favours a good match of the simulated ice thickness with respect to observations but it does not include any constraints on the ice velocity. It is thus not particularly

surprising that GRISLI performs better in terms of ice thickness than in terms of ice velocity. Since ice velocity is a very heterogeneous variable, it is sometimes convenient to use the logarithm of the velocity instead of the absolute velocity. When using the logarithm of the velocity GRISLI slightly improves compared to the other participating models since the RMSE is about 0.55 log(velocity in m yr$^{-1}$) (eleventh worst value out of 21). This means that the errors are mostly localised in areas of high velocities. Fig. 3a shows the absolute simulated velocity, to be compared to the observations in Fig. 3b. The pattern

is generally well reproduced and the model is able to reproduce the localisation of the major existing ice streams. However, the velocity of the ice streams is not always in agreement with the observational data. The northern and western ice streams are generally too slow with an underestimation reaching more than 500 m yr$^{-1}$ for the Jakobshavn, Petermann and North East Greenland ice stream (NEGIS) glacier termini (Fig. 3c). On the contrary, the south-eastern glaciers, Kangerdlugssuaq and Helheim, are too fast in the model. For the northern and western regions, the errors in ice thickness are small, meaning that

the ice velocity mismatch there cannot be reduced within our initialisation procedure which only minimises the ice thickness error. This is somewhat different for the south-eastern region, where there are important errors in ice thickness. However, there is an important positive bias in ice thickness at the ice sheet margin that tends to produce very high ice flow (very low basal drag coefficient to reduce this bias). Since the SSA equation is elliptic, the low basal drag at the margin has a regional impact on ice flow, which tends to produce an underestimation of the ice thickness further inland. While we strongly overestimate the

velocity in this area, the ice thickness at the margin is still overestimated. This suggests that the surface mass balance used in our reference climate is probably overestimated in this region.

The ice sheet model drift can be assessed by examining Fig. 2c. The ice thickness drift in the control experiment *ctrl_proj* is generally very small (lower than 10 metres) with only a few regions with higher values. Here again, the Kangerdlugssuaq and

Helheim glacier regions show the largest model drift with a local increase in ice thickness of more than 100 metres near the glacier termini. Overall the ice mass drift is negligible over the duration of the control experiment (86 years), also because of some compensating biases (see also Fig. 4). In addition to the ice thickness drift, the model simulates a drift in velocities during the duration of the control experiment (Fig. 3d). For most of the ice sheet the velocity change is small and only reaches more than 1 m yr$^{-1}$ at the ice sheet margins. The largest changes concern the glaciers in South-East Greenland such as the Helheim

and the Kangerdlugssuaq glaciers where locally, at the termini, there can be an increase in velocity by more than 1000 m yr$^{-1}$.

### 3.2 Ice sheet evolution projections

### 3.2.1 Sensitivity to climate forcing

Amongst the different experiments, we start with the description of the simulated ice sheet evolution under the RCP8.5 scenario for the 6 available CMIP5 models (Tier 1 and Tier 2). The simulated total ice mass evolution over the 1995-2100 period is

shown in Fig. 4 (expressed in total mass and in contribution to global sea level rise). In 2100, the total ice loss ranges from about -15 to -35$\times 10^3$Gt. This translates to a Greenland ice sheet melt contribution to global sea level rise of 35 mm of sea level equivalent (mm SLE) to 80 mm SLE. The 2100 sea level contribution simulated by GRISLI is close to the mean model response amongst the ISMIP6 participating models (Goelzer et al., 2020). The spread amongst the different climate forcings of about 20$\times 10^3$Gt (or 45 mm SLE) is thus larger than the mass change yielded with the GCM providing the smallest ice

sheet response (CSIRO-Mk3.6). The evolution of ice loss over the 86 simulated years is not linear, with an acceleration for all climatic scenarios. However, we can not discern any sudden change in the total mass evolution over the next century that may indicate a tipping point. The differences in mass evolution are tightly linked to the surface mass balance evolution for the different climate forcings. Amongst the CMIP5 climate models, IPSL-CM5-MR and MIROC5 simulate a mean surface mass balance becoming negative as early as 2060, while it remains positive over the next century for CSIRO-Mk3.6 (Fig. 5).

The CMIP6 models used in ISMIP6-Greenland have an Earth climate sensitivity from 4.8 to 5.3 °K, i.e. larger than the CMIP5 models used here, which show a range from 2.7 to 4.6 °K (Meehl et al., 2020). This has important consequences on the projected Greenland ice sheet. The total ice mass evolution for the four CMIP6 models under the SSP585 scenario is shown in Fig. 6. The CMIP6 models produce systematically higher ice loss than the CMIP5 models. The two most sensitive CMIP6 mod-

els (UKESM1-CM6 and CESM2) almost double the ice loss with respect to the most sensitive CMIP5 models (IPSL-CM5-MR and MIROC5). The ice loss thus reaches -60$\times 10^3$Gt (140 mm SLE) by the end of the century. This has also been reported by Greve et al. (2020) where the use of the CMIP6 model ensemble under the SSP585 leads to an ice sheet contribution to sea level rise increased by at least 70 % with respect to the contributions simulated using the CMIP5 ensemble.

Two climate models have been run under two scenarios for the evolution of future atmospheric greenhouse gases. The ice loss for the two scenarios of the climate models is shown in Fig. 7. The CMIP5 (MIROC5) and CMIP6 (CNRM-CM6) responses to the change in greenhouse gas scenario (RCP8.5 to RCP2.6 and SSP585 to SSP126 respectively) is very similar. There are very small differences for the first half of the century but after 2060 the high emission scenario produces substantial additional mass loss compared to the low emission scenario. By the end of the century, the high emission scenario produces roughly

-25$\times 10^3$Gt (55 mm SLE) of additional ice loss compared to the low emission scenario. The future atmospheric and oceanic warming induced by the greenhouse gas mixing ratio is thus a major driver for the Greenland ice mass loss at the century time scale.

The spatial pattern of ice loss by the end of this century is shown in Fig. 8. For this figure we have chosen four projection experiments that show contrasted integrated ice mass loss by 2100: the CISRO-Mk3.6 under RCP8.5 which produces a small integrated ice loss (Fig. 8a), the MIROC5 under RCP8.5 which produces an important mass loss (Fig. 8b), the MIROC5 under RCP2.6 to show the impact of the low emission scenario (Fig. 8c) and UKESM-CM6 under SSP585 with a high oceanic

sensitivity which produces the highest mass loss amongst all the different experiments (Fig. 8d). While the amplitude of ice thickness change is drastically different amongst these experiments, the spatial pattern is similar. The major signal is a substantial widespread ice thickness decrease at the margin of the ice sheet. If the ice thickness decrease is about 50 m for the least sensitive model (CSIR-Mk3.6), it can reaches more than 200 m for the most sensitive model (MIROC5 or IPSL-CM5-MR). The south-western region shows the largest ice sheet thinning. On the contrary, the central region shows a slight increase in ice

thickness which can reach about 50 m at places for the most sensitive climate scenario. This increase in ice thickness is related to the slight increase in precipitation simulated by some GCMs in the course of the century. The central eastern region shows only limited ice thickness changes regardless of the climate forcing used. The use of the RCP2.6 emission scenario reduces drastically the ice thickness changes.

### 3.2.2 Importance of the oceanic forcing

The uncertainty that arises from the oceanic forcing can be evaluated thanks to the different glacier retreat scenarios (*low*, *medium* and *high* sensitivity to oceanic forcing). In Fig. 4 is represented on the right-hand side the uncertainty that arises from the oceanic forcing for the individual CMIP5 models. In 2100 the ice mass loss difference between the *low* and *high* oceanic sensitivities is generally of about $-5 \times 10^3$Gt (less than 10 mm SLE). Without being negligible, the oceanic sensitivity for a given

climate scenario is nonetheless relatively small compared to the spread amongst the different CMIP5 climate models used. For the CMIP6 experiments, the uncertainty that comes from the oceanic forcing is almost doubled with respect to the CMIP5 experiments, with about $10 \times 10^3$Gt (~20 mm SLE) of ice loss difference from the *low* to *high* oceanic sensitivity (Fig. 6) but these CMIP6 models also produce much greater ice loss.

We also performed experiments in which we isolate the response of the model that arises from the atmospheric forcing only (first subset of Tier 3). For the *atmosphere only* (AO) experiments, we do not impose a retreat rate for the outlet glaciers and only the atmospheric perturbation is taken into account. Conversely, for the *ocean only* (OO) experiments, there is no atmospheric perturbation (as in the control *ctrl_hist* experiment) but we do impose a retreat rate for the outlet glaciers. The ice mass evolution for these experiments is shown in Fig. 9. The OO experiments produce almost identical mass evolutions amongst

the different GCMs. This means that even if the glacier retreat is subject to uncertainties, with the methodology of Slater et al. (2019) it is nonetheless only weakly sensitive to the differences in the climate forcing used to elaborate it. Fig. 9 also shows that the atmospheric forcing is the main driver for ice loss for the GCMs that produce an important ice loss. Also, the sum of the ice loss of AO and OO experiments approximates closely the ice loss simulated when using the combined forcing (92 to

94% of the combined forcing).

### 3.2.3 Change in ice dynamics

Climate forcing, and its associated ice sheet geometry change, leads to a change in the dynamics of the Greenland ice sheet. Fig. 10a shows the change in the simulated surface velocities at the end of the century with respect to the year 2015 for a given climate forcing. On the one hand, consistently with what has been found in previous studies (e.g. Peano et al., 2017; Le clec'h et al., 2019a), there is a decrease in simulated velocities related to ice thinning at the margins. On the other hand, the increase in surface slopes due to ice thinning at the margin leads to increased velocities further upstream.

The change in ice dynamics can also be assessed by investigating the different terms of the mass conservation equation. The integration in time of Eq. 1 over 2015-2100 suggests that the integrated ice flux convergence is the difference between the ice thickness change from 2015 to 2100 and the integrated mass balance (surface and basal mass balance and calving) over this period. The integrated divergence of the ice flux can be considered as the dynamical contribution to ice thickness change. It should be noted that the integrated mass balance here also includes the effect of ice mask change and surface elevation change. As such, it is not comparable to what would have been obtained with an atmospheric model only. Fig. 10b shows the difference of the dynamical contribution in 2100 for a selected climate forcing with respect to the control *ctrl_proj* experiment. The pattern mostly follows the one of velocity change (Fig. 10a). There is an important positive dynamical contribution to ice thickness change at the margins that tends to partially compensate the decrease in surface mass balance. Conversely, upstream regions show a slightly negative dynamical contribution. This pattern is similar amongst the different climate forcings. To compare the relative importance of the dynamical contribution with respect to surface mass balance to explain the ice thickness change, we show the ice thickness change in 2100 with a similar colour scale (opposite values and invert colour gradient) in Fig. 10c. The dynamical contribution shows generally much smaller values, suggesting that surface mass balance explains the largest changes in ice thickness. However, locally, for example in the south-eastern and central western regions, the dynamical contribution can be the largest driver of ice thickness change.

The inferred basal drag coefficient during the initialisation procedure is left unchanged for the duration of the historical and projection experiments. This is probably an important and inaccurate approximation since the basal conditions are susceptible to respond to changes in ice geometry and, eventually, basal hydrology. To assess the importance of basal drag coefficient changes for our projections, we perform a new set of experiments using the MIROC5 climate forcing under RCP8.5 with a medium oceanic sensitivity. For these simulations, we apply a spatially uniform modification factor to reduce or increase the value of the basal drag coefficient after the year 2045. The modification ranges from -90% (reduction to 10% of the initial value) to +100% (doubling of the initial value). The total ice mass difference in 2100 with respect to the experiment with no modification of the basal drag coefficient is shown in Fig. 11a,b. The mass change in response to small perturbations of the basal drag coefficient is relatively linear and limited. Thus, a perturbation of 20% results in less than $5\times10^3$Gt mass change,

which translates to less than 10 mm SLE. This means that it is unlikely that basal condition changes in the future could produce a drastically different total ice mass change in 2100. This also suggests that a slightly different basal drag coefficient inferred during our initialisation procedure will produce a similar mass evolution in the projection experiments. In order to further assess the sensitivity of our projections to the choice of mechanical parameters, we repeated these perturbation experiments for the SIA flow enhancement factor (Fig. 11c,d). We varied the enhancement factor from 0.4 to 6 with respect to the standard value of 1. As for the basal drag coefficient perturbation, the response in terms of ice mass loss is small and relatively linear.

To assess the range of acceptable values for the basal drag coefficient perturbation and the enhancement factor, we also performed similar sensitivity experiments for the control experiment *ctrl_proj*. The range of acceptable perturbations is thus defined as the perturbed control experiments that produce less than 0.1% total mass change with respect to the standard control experiment. 0.1% total mass change corresponds to one tenth of the total mass change in 2100 with respect to 2015 using the MIROC5 climate forcing under RCP8.5 with a medium oceanic sensitivity. The acceptable perturbations of the basal drag coefficient range from -15% to 20% and the acceptable enhancement factors range from 0.8 to 1.2. Interestingly, the effect of the perturbations (basal drag coefficient or enhancement factor) on the mass change is almost identical for the projection experiments (blue dots in Fig. 11) and for the control experiments (light blue dots in Fig. 11). This means that, in our model, different mechanical parameters do not enhance nor mitigate the mass loss due to climate change.

## 4   Discussion

In order to minimise the initial error in ice thickness with respect to the observations, we have used an inverse procedure that optimally tunes the basal drag coefficient. In doing so, we produce a simulated ice sheet that is in quasi-equilibrium with the climate forcing (minimal ice thickness drift). In reality, the Greenland ice sheet is far from being at equilibrium with the present-day climate since it has been losing mass at an accelerated rate over the last four decades (Mouginot et al., 2019). This means that, by construction, our simulations underestimate the Greenland ice sheet contribution to future sea level rise. A simple linear extrapolation of the 2006-2016 rate (0.77 mm yr$^{-1}$,  Oppenheimer et al., 2019) up to 2100 would result in a 6.5 cm SLE from the Greenland ice sheet. This number is comparable to the GRISLI spread discussed in this paper, and more generally to the spread amongst ISMIP6 models (3.5 to 14 cm SLE,  Goelzer et al., 2020). This suggests that model initialisation is one of the largest sources of uncertainty for model projections. Instead of using a methodology that produces an ice sheet at equilibrium, some promising alternatives exist, for example using data assimilation of observed velocities in a transient ice sheet simulation (Gillet-Chaulet, 2020). These methods require however a complex data assimilation framework, currently not implemented in our ice sheet model. Instead, we plan to modify the inverse procedure of Le clec'h et al. (2019b) by incorporating the ice thickness change inferred by gravimetry/altimetry as an additional constraint in order to improve on the initial state of the Greenland ice sheet.

One additional limitation of the inverse procedure is that it does not take into account the impact of the last glacial cycle on ice temperatures. Our internal temperature field is the result of a long thermo-mechanical equilibrium under perpetual present-day forcing and as such, it is necessarily overestimated since the memory of the low temperatures of the glacial period in the ice sheet is not accounted for. In addition to an underestimated ice viscosity, this has also consequences on the simulated basal tem-

perature and, as a result, on the regions where sliding occurs. This might affect the dynamical response of the model to future climate change. If earlier studies have already identified these limitations (e.g. Rogozhina et al., 2011; Yan et al., 2013; Seroussi et al., 2013; Le clec'h et al., 2019b), our inverse procedure does not allow for a quantification of these limitations. Given the relatively low computational cost of GRISLI, one alternative would be to perform muti-millenial palaeo integrations to infer the initial state used for the projections. This strategy generally leads to a larger ice thickness error with respect to present-day

observations but has the advantage to have a thermal state consistent with the model physics and with the palaeo temperatures. While the ISMIP6-Greenland participating models either choose one or the other initialisation technique (Goelzer et al., 2020), it would be very informative to have two drastically different initialisation methods for a given ice sheet model.

The forcing methodology used for ISMIP6-Greenland accounts for the vertical elevation feedback on temperature and surface

mass balance. In order to quantify the impact of this correction on the simulated evolution of the ice sheet, we run a sensitivity experiment in which this correction is not accounted for. Using MIROC5 under the RCP8.5 scenario with a medium oceanic sensitivity, we simulate a Greenland contribution to future sea level rise 5.1% smaller compared to the same experiment in which the vertical correction is applied. This number is slightly higher than the effect reported by Edwards et al. (2014) and Le clec'h et al. (2019a) (4.3 and 4.2 % respectively), but smaller than that of Vizcaino et al. (2015) (8-11%) and Calov et al.

(2018) (about 13%). Differences in resolution and/or physical processes implemented in the atmospheric model could explain this diversity.

In addition to the vertical elevation feedback on surface mass balance, other feedbacks at play for the future evolution of the Greenland ice sheet are not accounted for in the ISMIP6-Greenland methodology. Notably, the MAR model used to compute

the forcing fields does not account for topography and ice mask changes. The effect of these changes is probably limited for moderate ice sheet retreat (Le clec'h et al., 2019a). However, since the CMIP6 models used here produce a much greater retreat than the CMIP5 models, they could also induce more important feedbacks if MAR was bi-directionally coupled to an ice sheet model. In addition, the effect of Greenland ice loss on the ocean is also not taken into account with the forcing methodology followed. While the oceanic forcing seems not to be the major driver for future Greenland ice loss, glacier retreat in the future

should ideally take into account the oceanic circulation changes in the fjords related to freshwater discharge from ice sheet melting.

## 5   Conclusions

In this paper we have presented the GRISLI-LSCE contribution to ISMIP6-Greenland. Independently from the climate forcing used to drive the ice sheet model, we have shown that the Greenland ice sheet systematically loses ice in the future. However, the magnitude of the mass loss by 2100 is very sensitive to the climate forcing. Under a *business as usual* scenario for the greenhouse gas emission (RCP8.5 or SSP585), the mass loss translates into a Greenland ice sheet contribution to global sea level rise that ranges from 35 to 160 mm SLE. However, with a low emission scenario for greenhouse gases (RCP2.6 or SSP126) the mass loss can be significantly reduced. The CMIP6 models selected for ISMIP6 tend to produce a larger ice loss due to their higher climate sensitivity with respect to the one of the CMIP5 models. The oceanic forcing contributes to ice loss by about 10 mm SLE in 2100. In addition, the time integral of the surface mass balance is generally much larger than the dynamical contribution to ice thickness change (by an order of magnitude). This suggests that the Greenland ice mass loss in the future is mostly driven by surface mass balance changes, in particular through a larger ablation at the ice sheet margin. This process should thus be carefully implemented in ice sheet models aiming at simulating the Greenland ice sheet evolution at the century scale. With additional sensitivity experiments, not included in ISMIP6, we have also shown that the choice of uncertain mechanical parameters (i.e. flow enhancement factor and basal drag coefficient) has only a small impact on the spread of mass loss. Finally, the initial condition chosen for the ice sheet model remains an important question for ice sheet modelling. In particular, assuming an ice sheet in equilibrium with present-day climate for the initial condition, as done here but also in most ISMIP6 participating models, could lead to an underestimation of the future mass loss.

## 6   Data availability

The GRISLI outputs from the experiments described in this paper are available on the Zenodo repository with digital object identifier 10.5281/zenodo.3784665 (Quiquet and Dumas, 2020b). The outputs in the Zenodo repository are the standard GRISLI outputs on the native 5 km grid and, as a result, they may slightly differ from the post-processed outputs available on the official CMIP6 archive on the Earth System Grid Federation (ESGF). In order to document CMIP6's scientific impact and enable ongoing support of CMIP, users are obligated to acknowledge CMIP6, the participating modelling groups, and the ESGF centres (see details on the CMIP Panel website at http://www.wcrp-climate.org/index.php/wgcm-cmip/about-cmip). The forcing datasets are available through the ISMIP6 wiki (http://www.climate-cryosphere.org/wiki/index.php?title=ISMIP6_wiki_page, last access: 13 January 2021).

*Acknowledgements.* We thank the Climate and Cryosphere (CliC) effort, which provided support for ISMIP6 through sponsoring of workshops, hosting the ISMIP6 website and wiki, and promoted ISMIP6. We acknowledge the World Climate Research Programme, which, through its Working Group on Coupled Modelling, coordinated and promoted CMIP5 and CMIP6. We thank the climate modeling groups for producing and making available their model output, the Earth System Grid Federation (ESGF) for archiving the CMIP data and providing access, the University at Buffalo for ISMIP6 data distribution and upload, and the multiple funding agencies who support CMIP5 and

CMIP6 and ESGF. We thank the ISMIP6 steering committee, the ISMIP6 model selection group and ISMIP6 dataset preparation group for their continuous engagement in defining ISMIP6. This is ISMIP6 contribution No 23.

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

**Table 1.** List of ISMIP6-Greenland experiments performed in this work.

| exp_id | scenario | GCM | Ocean | |
|---|---|---|---|---|
| exp05 | RCP8.5 | MIROC5 | Medium | |
| exp06 | RCP8.5 | NorESM | Medium | |
| exp07 | RCP2.6 | MIROC5 | Medium | Core experiments – Tier 1 |
| exp08 | RCP8.5 | HadGEM2-ES | Medium | |
| exp09 | RCP8.5 | MIROC5 | High | |
| exp10 | RCP8.5 | MIROC5 | Low | |
| expa01 | RCP8.5 | IPSL-CM5-MR | Medium | Extended ensemble – Tier 2 |
| expa02 | RCP8.5 | CSIRO-Mk3.6 | Medium | |
| expa03 | RCP8.5 | ACCESS1.3 | Medium | |
| expb01 | SSP585 | CNRM-CM6 | Medium | |
| expb02 | SSP126 | CNRM-CM6 | Medium | CMIP6 extension – Tier 2 |
| expb03 | SSP585 | UKESM1-CM6 | Medium | |
| expb04 | SSP585 | CESM2 | Medium | |
| expb05 | SSP585 | CNRM-ESM2 | Medium | |
| expc01 | RCP8.5 | MIROC5 AO | Medium | |
| expc02 | RCP8.5 | MIROC5 OO | Medium | |
| expc03 | RCP8.5 | CSIRO-Mk3.6 AO | Medium | |
| expc04 | RCP8.5 | CSIRO-Mk3.6 OO | Medium | Ocean only (OO) and Atmos. only (AO) – Tier 3 |
| expc05 | RCP2.6 | MIROC5 AO | Medium | |
| expc06 | RCP2.6 | MIROC5 OO | Medium | |
| expc07 | RCP8.5 | NorESM AO | Medium | |
| expc08 | RCP8.5 | NorESM OO | Medium | |
| expc09 | RCP8.5 | MIROC5 AO | Low | |
| expc10 | RCP8.5 | MIROC5 OO | High | |

| exp_id | scenario | GCM | Ocean | |
|---|---|---|---|---|
| expd01 | RCP8.5 | NorESM | High | |
| expd02 | RCP8.5 | NorESM | Low | |
| expd03 | RCP8.5 | HadGEM2-ES | High | |
| expd04 | RCP8.5 | HadGEM2-ES | Low | |
| expd05 | RCP8.5 | MIROC5 | High | |
| expd06 | RCP8.5 | MIROC5 | Low | |
| expd07 | RCP8.5 | IPSL-CM5-MR | High | |
| expd08 | RCP8.5 | IPSL-CM5-MR | Low | |
| expd09 | RCP8.5 | CSIRO-Mk3.6 | High | |
| expd10 | RCP8.5 | CSIRO-Mk3.6 | Low | |
| expd11 | RCP8.5 | ACCESS1.3 | High | Ocean sensitivity – Tier 3 |
| expd12 | RCP8.5 | ACCESS1.3 | Low | |
| expd13 | SSP585 | CNRM-CM6 | High | |
| expd14 | SSP585 | CNRM-CM6 | Low | |
| expd15 | SSP126 | CNRM-CM6 | High | |
| expd16 | SSP126 | CNRM-CM6 | Low | |
| expd17 | SSP585 | UKESM-CM6 | High | |
| expd18 | SSP585 | UKESM-CM6 | Low | |
| expd19 | SSP585 | CESM2 | High | |
| expd20 | SSP585 | CESM2 | Low | |
| expd21 | SSP585 | CNRM-ESM2 | High | |
| expd22 | SSP585 | CNRM-ESM2 | Low | |

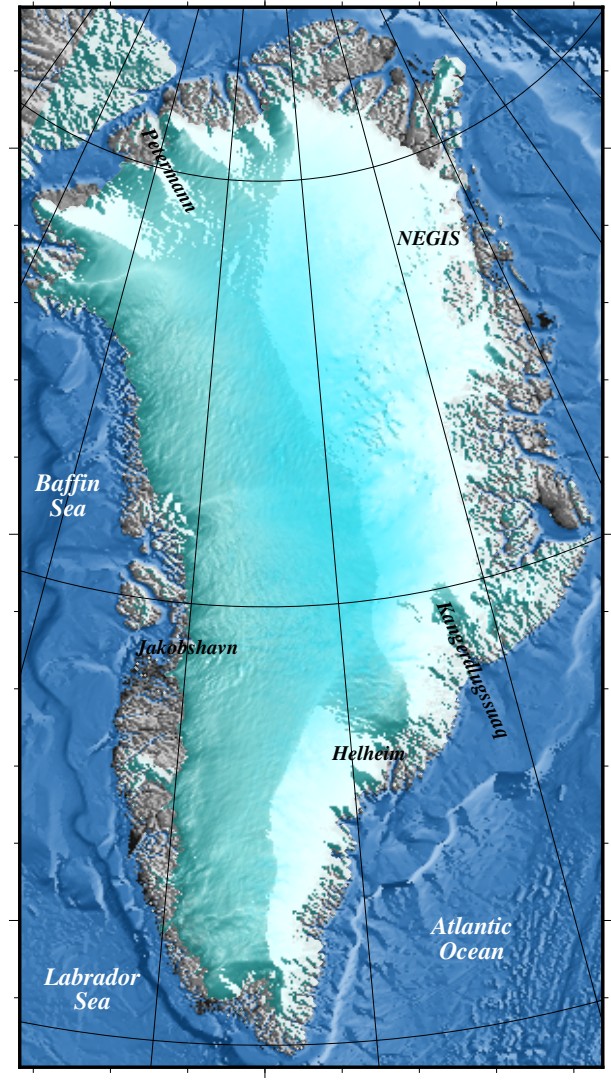

**Figure 1.** The Greenland ice sheet with the major ice streams discussed in the text.

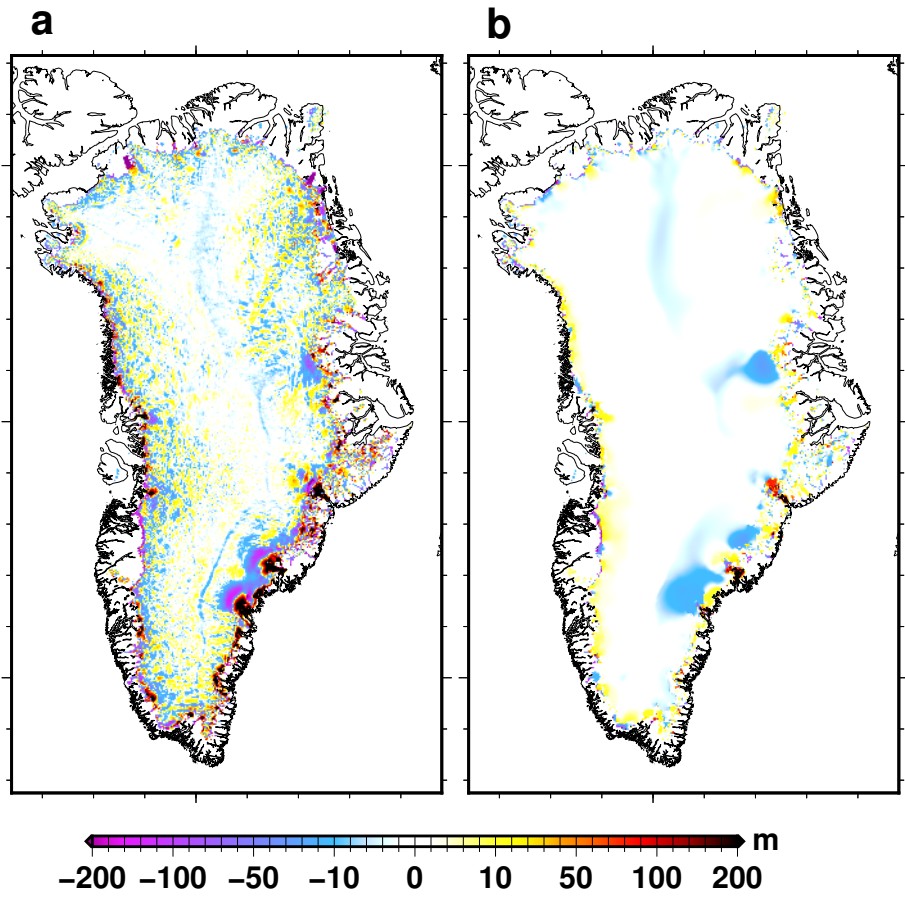

**Figure 2.** Ice thickness difference: **(a)** end of the historical experiment *hist* (2015) with respect to the observations (Morlighem et al., 2017); **(b)** end of the control experiment *ctrl_proj* (2100) with respect to the end of historical experiment *hist* (2015).

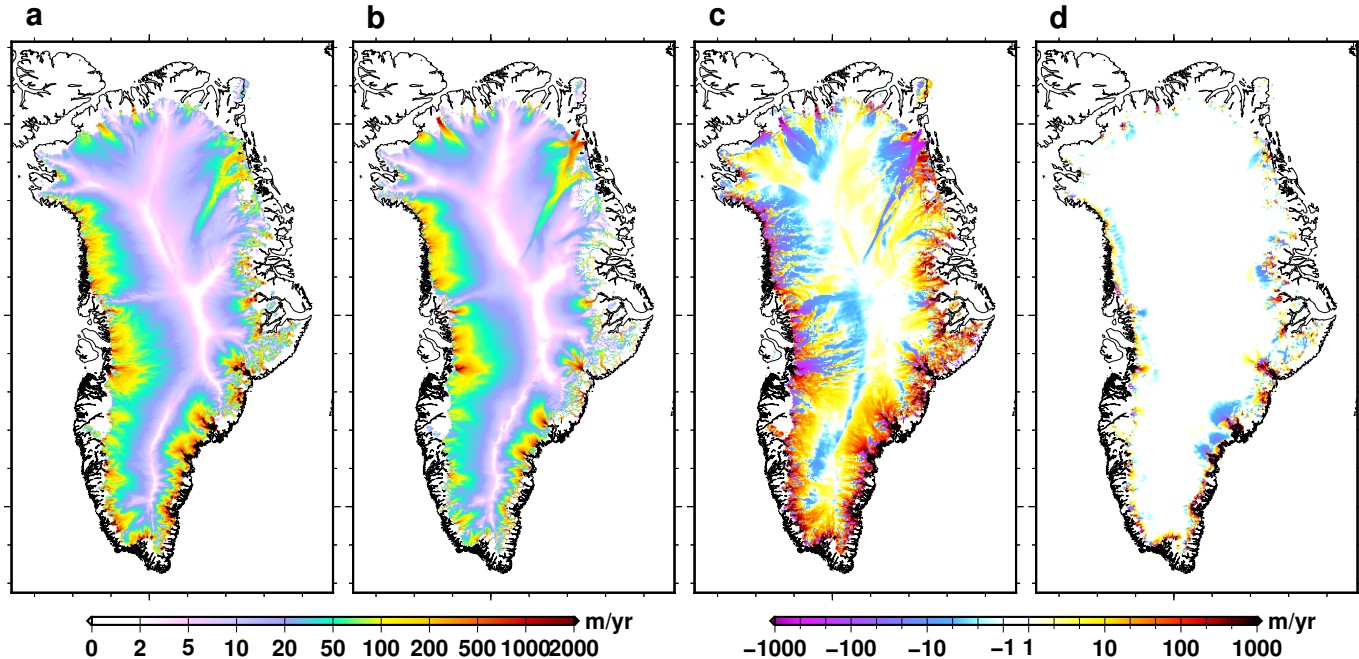

**Figure 3.** Surface velocity magnitude: **(a)** simulated at the end (2011-2015) of the historical experiment *hist*; **(b)** in the observational datasets of Joughin et al. (2016); **(c)** difference between (a) and (b). The surface velocity magnitude change from 2011-2015 to 2096-2100 in the control experiment *ctrl_proj* is shown in **d**. We use a 5 year mean for the simulated velocity to reduce the impact of interannual variability. The range -1 to 1 m yr$^{-1}$ is set to white for the velocity differences (**c** and **d**).

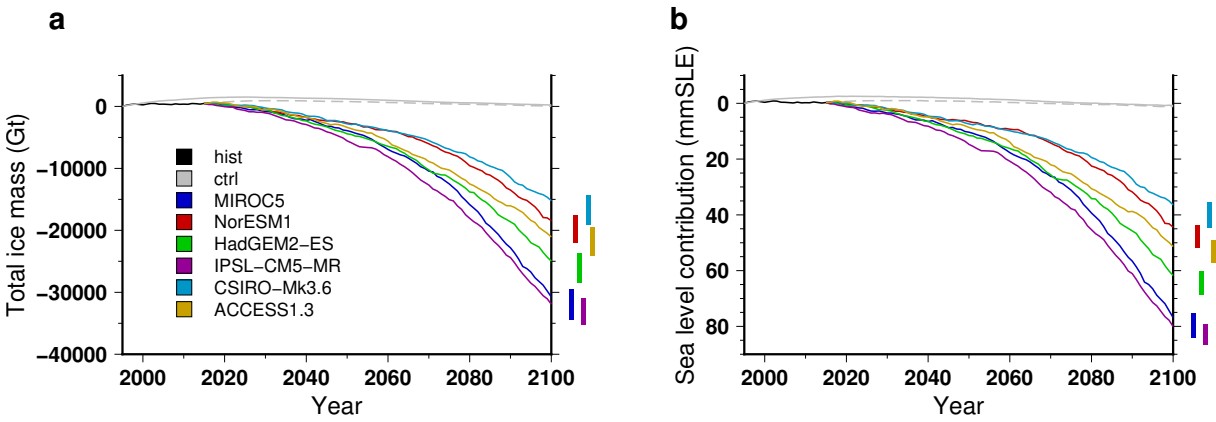

**Figure 4.** Simulated total ice mass change for the historical simulation *hist* (1995-2015), the control experiments *ctrl* (solid grey lines) and *ctrl_proj* (dashed grey lines) and the projections under the different CMIP5 forcings using the RCP8.5 scenario and the medium oceanic sensitivity: **(a)** total mass change and **(b)** ice volume contributing to sea level rise. For each projection experiment the right-hand side vertical bar shows the minimal and maximal changes associated with the oceanic forcing uncertainty (*low* and *high* scenarios).

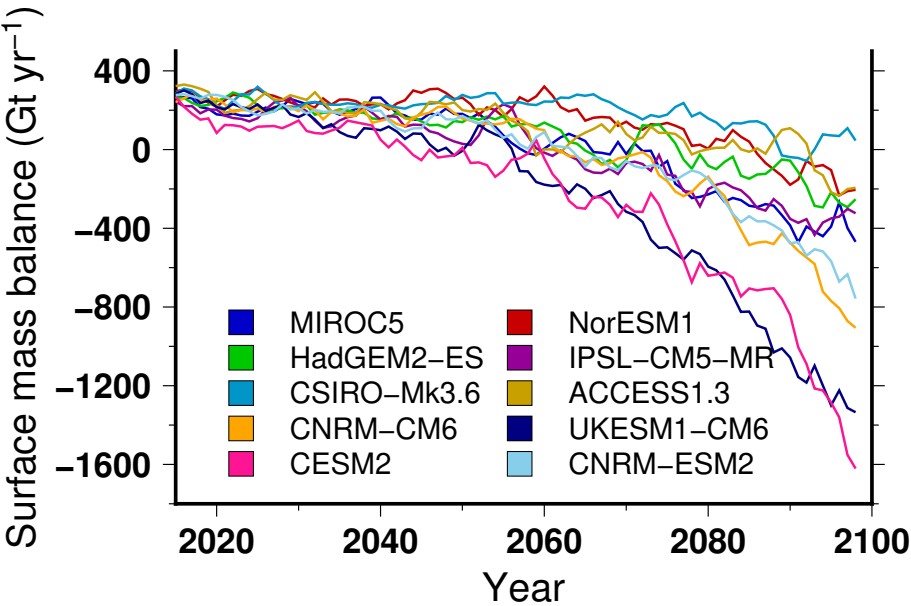

**Figure 5.** Simulated surface mass balance, integrated over the ice sheet, for different CMIP5 and CMIP6 climate forcings using the RCP8.5 scenario and SSP585 scenario, respectively. The projection experiments shown in this figure use the medium oceanic sensitivity. For this figure we use a 5-year running mean in order to smooth the interannual variability.

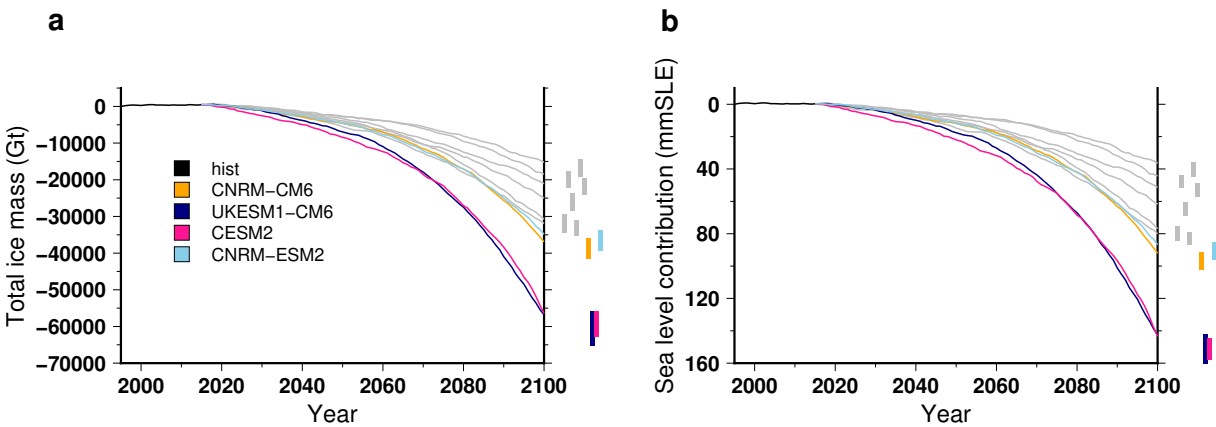

**Figure 6.** Simulated total ice mass change for the historical experiment *hist* (1995-2015) and the projections under the different CMIP6 forcings using the SSP585 scenario and the medium oceanic sensitivity: **(a)** total mass change and **(b)** ice volume contributing to sea level rise. For each projection experiment the right-hand side vertical bar shows the minimal and maximal changes associated with the oceanic forcing uncertainty (*low* and *high* scenarios). The grey lines are the changes under the CMIP5 forcings shown in Fig. 4.

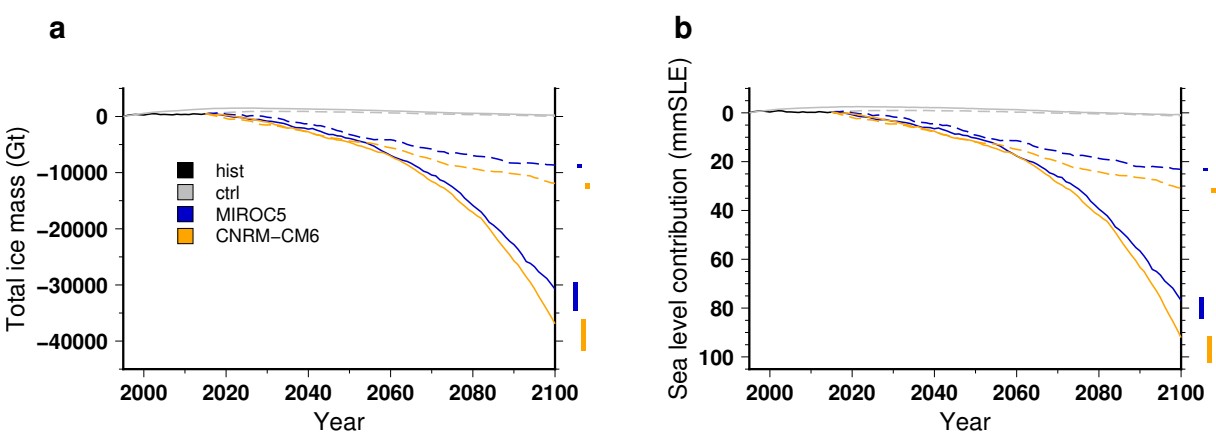

**Figure 7.** Simulated total ice mass change for the historical experiment *hist* (1995-2015), the control experiments *ctrl* (solid grey lines) and *ctrl_proj* (dashed grey lines) and for the projections using two climate models run under a high emission scenario for greenhouse gases (solid lines, RCP8.5 for MIROC5 and SSP585 for CNRM-CM6) and a low emission scenario (dashed lines, RCP2.6 for MIROC5 and SSP126 for CNRM-CM6) with a medium oceanic sensitivity, expressed as: **(a)** total mass change and **(b)** ice volume contributing to sea level rise. For each projection experiment, the right-hand side vertical bar shows the minimal and maximal changes associated with the oceanic forcing uncertainty (*low* and *high* scenarios)

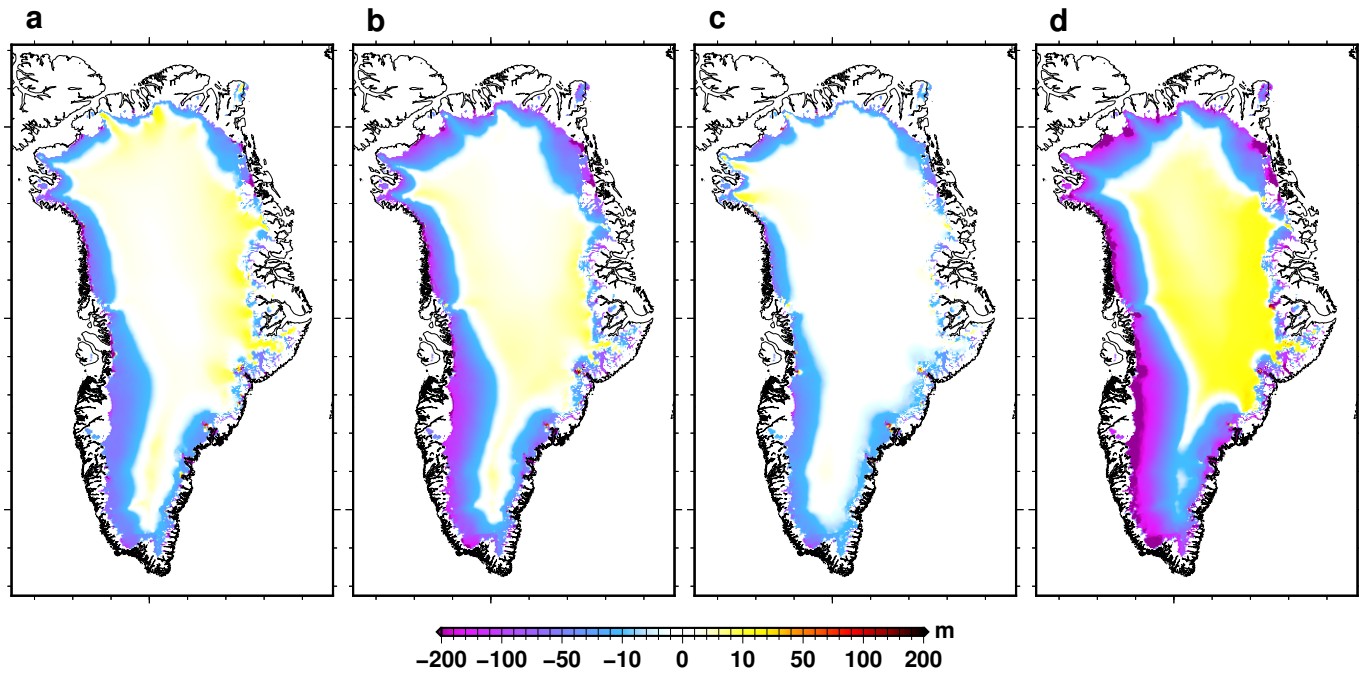

**Figure 8.** Simulated ice thickness change (2100 - 2015) for: **(a)** CSIRO-Mk3.6 (RCP8.5); **(b)** MIROC5 (RCP8.5); **(c)** MIROC5 (RCP2.6) and; **(d)** UKESM-CM6 (SSP585) climate forcing. The medium oceanic sensitivity has been used for this figure, except for UKESM-CM6 (d) for which we use the high oceanic sensitivity. The ice thickness change shown here is corrected for the ice thickness change (2100-2015) in the control experiment *ctrl_proj*.

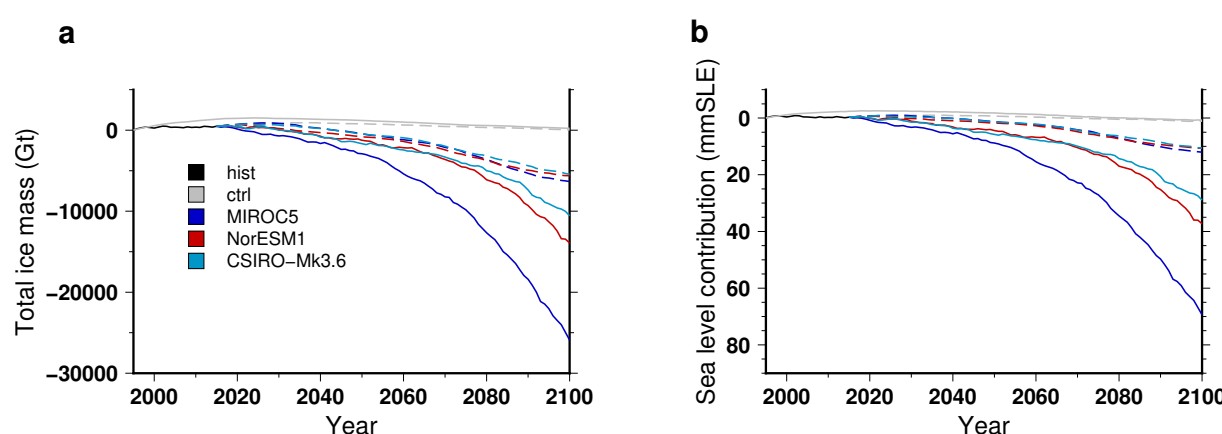

**Figure 9.** Simulated total ice mass change for the historical experiment *hist* (1995-2015), the control experiments *ctrl* (solid grey lines) and *ctrl_proj* (dashed grey lines) and the projections under different CMIP5 forcings using the RCP8.5 scenario. For the projections, the solid lines stand for experiments under atmospheric forcing change only (no imposed outlet glacier retreat, AO) while the dashed lines stand for experiments under oceanic forcing change only (no change in surface mass balance, OO). The changes are expressed as: **(a)** total mass change and **(b)** ice volume contributing to sea level rise. The medium oceanic sensitivity has been used for the oceanic only experiments (OO).

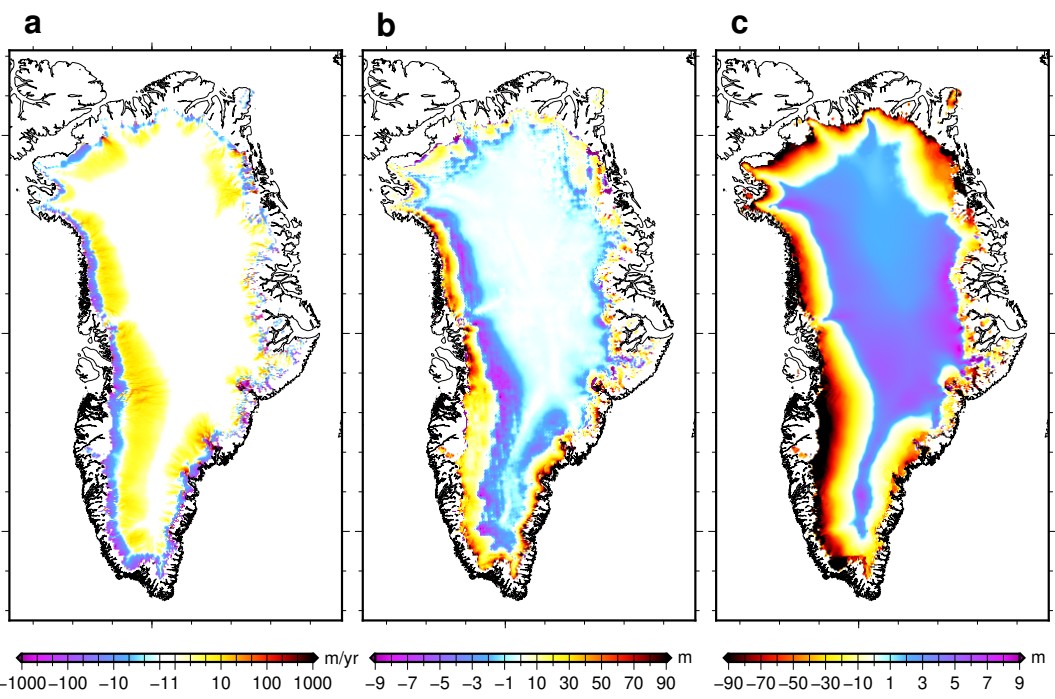

**Figure 10. (a)**: Simulated surface velocity change during the projection run (2096-2100 with respect to 2015-2019) using MIROC5 forcing under RCP8.5 with a medium oceanic sensitivity. **(b)**: change in the dynamical contribution to ice thickness change in 2100 (see text for definition) for this same experiment. **(c)**: simulated ice thickness change (2100-2015). For all panels, we corrected the changes by the ones simulated in the control experiment *ctrl_proj* over the same period. The range -1 to 1 m yr$^{-1}$ is set to white for velocity difference (**a**). The colour scale is not symmetrical for **(b)** and **(c)**.

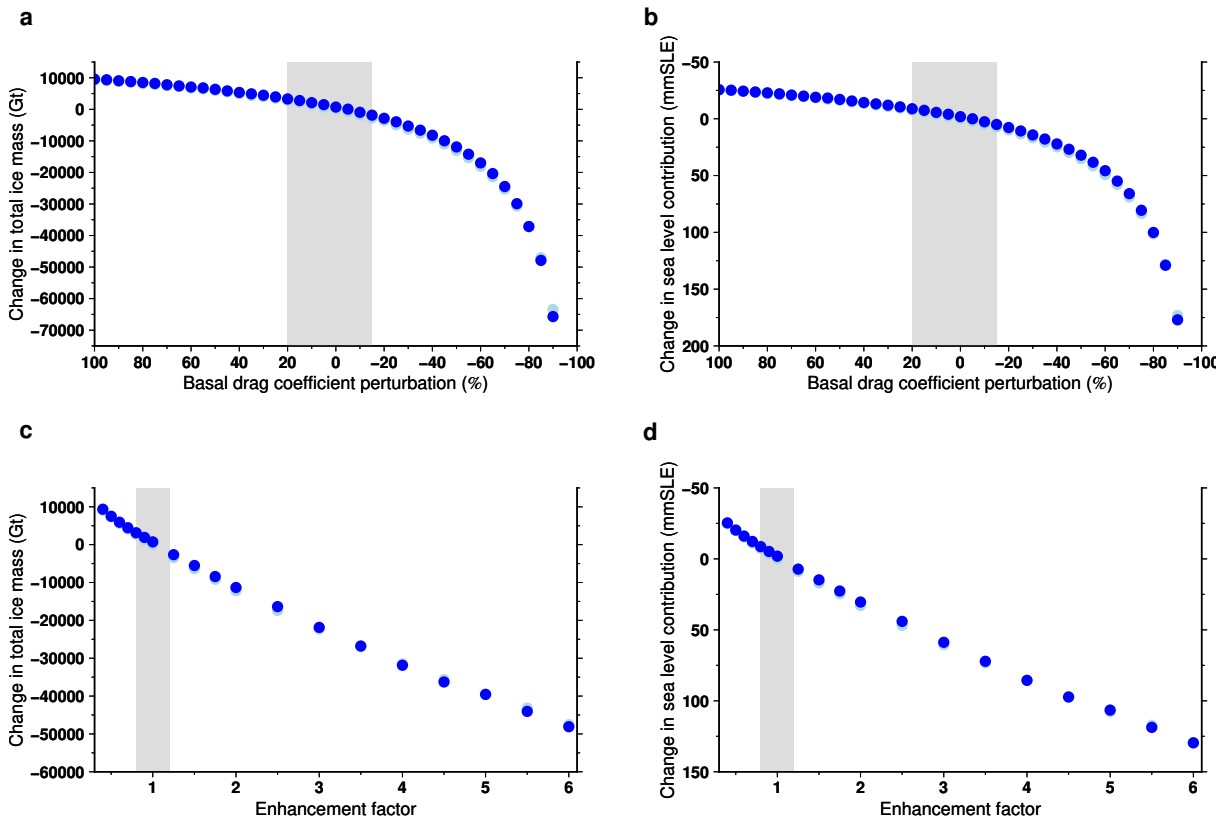

**Figure 11.** Change in ice volume for a modification of the basal drag coefficient (**a** and **b**) and different values of the enhancement factor (**c** and **d**). In this figure, each dot represents the difference in 2100 with respect to the standard experiment (no basal drag coefficient perturbation and enhancement factor at 1). The dark blue dots are projection experiments that use MIROC5 under RCP8.5 with a medium oceanic sensitivity. The light blue dots are control experiments *ctrl_proj*. Some control experiments can be hidden by the projection experiments if they imply a similar volume change. The perturbations are applied starting at year 2045. The vertical grey band stands for the range of perturbations that produce a 0.1% of total mass change in the perturbed control experiment with respect to the standard control experiment. The difference is expressed in total ice mass (**a** and **c**) and ice volume contributing to sea level rise (**b** and **d**).