# Peer review of "The GRISLI-LSCE contribution to ISMIP6, Part 1: projections of the Greenland ice sheet evolution by the end of the 21st century"

_The Cryosphere, 2020_

## Referee Comment (RC1) · Anonymous Referee #1 · 30 Jun 2020

This paper is clearly written and the figures are good. It describes the results of following the ISMIP6 Greenland experimental protocol with a particular dynamical ice-sheet model. Although this is information of use to assessing uncertainties in projections, the scientific gain is not clear. It would be useful if the authors could emphasise scientific lessons we learn from studying this model in particular, beyond its inclusion in the ISMIP6 comparisons, for example? Looking at the conclusions alone, I think a reader who is familiar with the literature of the last several years would find nothing new or surprising, for instance. However, in the paper there are a few new things which ISMIP6

is helping to clarify, and there are moreover useful things which have or could be done with this model, because it is computationally cheap, to test sensitivities.

A few of my comments relate to the importance of the SMB forcing, which the paper demonstrates. It would be useful to quantify (graphically or in numbers) how much of the spread among GCMs and scenario is due simply to the time-integral of the SMB forcing (as applied to the ice-sheet model), and not affected by the ice-sheet model itself. While it is certainly necessary to use a dynamical ice-sheet model to study large changes in ice-sheets, it would be useful if the authors could present evidence for the need to use one for the 21st century (when not coupled to the atmosphere or ocean), especially as doing so introduces complications of drift and spinup, as described by the paper.

I have some concern about the prescription of the large melting near the edge (o4 line 12) and the retreat masks (p4 line 32). With both of these enforced, is the dynamical behaviour of the model distorted?

p1 line 10-11. I would not jump to such a strong general conclusion (also on p7).

p1 line 17-18. I don't think that this statement (of a most likely contribution of 1 m from ice sheets by 2100) is a correct representation of the current state of scientific knowledge. In the first place, you can't state a likelihood independent of scenario, since there are no probabilities for scenarios. Bamber et al. write "For a +5degC temperature scenario, more consistent with unchecked emissions growth, the [median and 95-percentile] are 51 and 178 cm, respectively." I'm not sure what "most likely" means, but 1 m is twice their median. Also, Bamber et al. report an expert elicitation, whose reliability is debatable since it's opaque. For comparison, the AR5 assessment of the likely range of ice-sheet contributions by 2100 under RCP8.5 is 0.09 to 0.28 m from Greenland and -0.08 to 0.14 m from Antarctica.

p2 line 1. Why "asymptotic"?

p3 line 4. I suppose that strictly you could say an ice-sheet model satisfied momentum conservation, but as far as I know this model and others used for such purposes do not contain terms for acceleration or inertia. That is, momentum is always negligible, and they assume a balance of forces at all times.

p3 Eq 1. I think that BM is a single quantity, isn't it? Typeset like this in a formula it looks exactly like the product of two quantities B and M (like Ubar H is a product). It would be clearer to use a single symbol. Is it just the surface mass balance, or is basal mass balance included too?

p3 line 11. "the total velocity is simply to superposition of the two main approximation". I would suggest "the total velocity is the sum of the velocities predicted in their respective areas by the two main approximations".

p3 line 15. "for which there is infinite, respectively none, friction at the base." I think this should read "for which there is infinite friction at the base or none, respectively." "None" is a pronoun, not an adjective.

p4 line 5. How accurate are the SMB and the surface topography in the control state?

p4 line 11-14. Does this term strongly interfere with, or even overwhelm, the simulated discharge across the grounding line?

p4 line 24. State that these are vertical gradients. I would say that they are vertical gradients of surface quantities in the atmosphere model, rather than in the atmosphere.

p5 line 11. branched to -> branched from.

p4 line 21-22. What do you need the surface temperature for, if you're using SMB as forcing?

p5 line 22, p8 line 3, p11 line 11, Fig 5 caption. Although the reader may sympathise with the authors, it's better to avoid "pessimistic" and "optimistic", which are value-judgements.

p5 last para. I don't understand the reason for these two experiments. Do they start from the same initial state? Since they have the same forcing, they ought to evolve identically.

p5 line 34. alike -> like.

p6 line 22. best -> better.

p6 line 24. "In doing so" means doing what? - absolute or logarithm? I would have assumed logarithm, but the next sentence suggests otherwise. What are the units of 0.55? What are the units of velocity before taking the logarithm? (Strictly you can only take the log of a dimensionless quantity, but the conversion factor between different velocity scales will be a constant offset in the log so doesn't affect its RMSE, I suppose.)

p6 line 30. Why is this "on the contrary"? If I read this correctly, all the errors are in the same direction (too slow in the model). Can you suggest the reason for this systematic bias? What implication does it have for projections?

p7 line 4-5. What implication will this bias in SMB have for projections?

p7 line 9 and 15. Are these large drifts in thickness and velocity related? What effect will they have on projections? It's not obvious that you can simply subtract an unforced drift when it's large compared with the forced response.

p7 line 18. start by -> start with.

p7 line 21-24. Presumably this spread comes mostly from the spread in SMB forcing from the GCMs. Could you also add the ice-sheet area- and time-integral of the SMB perturbation to the graphs?

p7 line 21-24. It seems that these projections imply quite a low sensitivity to climate change compared with the models on which the AR5 was based; their assessment of the Greenland contribution by 2100 under RCP8.5 is 90-280 mm, of which 40-220 mm is from SMB change.

p7 line 25. What sort of "tipping point" do you have in mind, that you might see in the volume evolution? Can you give references to relevant suggestions?

p7 line 26-27. I think we should be more cautious in drawing conclusions. There are only four CMIP6 models considered in this study, out of dozens in total, and two of the four are at the edge of the CMIP5 distribution in your projections. Only two show much greater sensitivity, and those results are within the AR5 range.

p8 line 6-7. It's not the GHG itself which is the driver, but the warming it produces; that is also the reason why the rate of mass loss goes up with time, and the main reason for the spread among models.

p8 line 13-14. Since the point you wish to make is the similarity of the patterns, it would be better to show these maps divided by the integrated change in each case i.e. normalised to the same GMSLR contribution. That would reveal the patterns themselves, so they could be compared, which I agree should be the purpose of this figure.

p9 line 6. It would be interesting to see the time-integral of the applied SMB perturbation here, to compare with the AO experiments (as I also suggested on p7 for Fig 3). Any difference is due to the dynamical response to the SMB forcing.

p9 lines 16-23. The text says "Fig. 8b shows the difference in ice flux convergence in 2100", and the fig caption says "change in the dynamic contribution to ice thickness change in 2100". I don't think either of those is a correct description, if I have understood correctly. You also say, "This can be considered as the dynamical contribution to ice thickness change," which I think is correct. The quantity shown is the difference (change in topography during the experiment) minus (time-integral during the experiment of the local mass balance change with respect to control) - is that right? It would be useful to compare this difference with the change in topography in the same experiment, using the same color scale, in order to see the relative importance of the dynamical change. If it's a small fraction, you might argue that there's no need to use a dynamical ice-sheet model for projections on this timescale. Where it's not small, you

can comment. Part of the dynamical contribution near the coast is a response to the ocean forcing, I presume. Therefore it would also be useful to show the same comparison for the AO experiment. That is, would it be good enough to make the projection without a dynamical model, simply by time-integrating the local SMB perturbation?

p9 line 30. As a guide to the possible magnitude of this underestimate, you could state what the presently observed ice-sheet imbalance would give if it continued as a constant rate to 2100 and compare with your projected changes in response to forcing.

p10 line 4-16. This is useful, but it's not really discussion, I'd say. It's another sensitivity test, and it would go well in sect 3.2.3 about change in ice dynamics.

p10 line 20. Why is it necessarily an overestimate?

p10 line 27-29. Yes, it would! Since your model is particularly computationally inexpensive, please could you do it and tell us the answer? :-)

p10 line 31-32. Could you quantify the elevation-SMB feedback here, or earlier, and compare it with Edwards et al. (Cryosphere, 2014)? You could directly quantify it by running a sensitivity test in which the lapse-rate adjustment is excluded, I suppose.

p11 line 8. is systematically loosing -> systematically loses

Fig 1 caption. Does "respective to" mean "with respect to"? For clarify please state the years of the end of the historical and end of ctrl_proj.

---

## Referee Comment (RC2) · Anonymous Referee #2 · 16 Jul 2020

In this manuscript, the authors report on their ISMIP6 Greenland projections with the model GRISLI. The paper is easy and straightforward to follow. Its scientific value beyond the community publication (Goelzer et al., 2020, in press) lies in a more detailed description of the set-up of GRISLI, a more detailed analysis of the results and the fact that the entire suite of ISMIP6 experiments (Tier 1-3) are dealt with.

Overall, I found the results interesting and the presentation adequate. I'd only like to raise some issues that should be dealt with as follows:

[Figure]

The English writing clearly has some room for improvements. I am not going to point out all the issues, but just some examples from the first page: P. 1, l. 3/4: "an increase_d_ mass loss". P. 1, l. 5: "the largest single source contribution _after_ the thermosteric contribution". P. 1, l. 19/20: Assessment of projections? Either "need for assessment of future SLR by projections" or "need for projections of future SLR". P. 1, l. 22: Strange formulation: "from changing boundary conditions such as climate change". Before resubmission, the entire manuscript should be very carefully proof-read by a (near-) native speaker or a professional language editing service.

Throughout MS (e.g., p. 1, l. 10, l. 14): "mmSLE" -> "mm SLE"

Throughout MS (e.g., p. 2, l. 4): "in term of" -> "in terms of"

P. 1, l. 17/18, "most-likely amplitude exceeding 1 metre in 2100": This is not what has been found in the ISMIP6 ensemble projections (Goelzer et al., 2020, TC, in press; Seroussi et al., 2020, TC, in press). Even with the most sensitive model results, it is less than half a metre combined. At some point in the paper, this should be mentioned.

P. 3, l. 7, 17: Add commas after the displayed equations.

P. 3, l. 11-13: The description of the SIA and SSA is over-simplified. Starting from full Stokes, in both cases, some horizontal and some vertical derivatives of the components of the stress tensor are neglected. In very compact form, this is shown in the tutorial at http://doi.org/10.5281/zenodo.3739009, p. 22 (for SIA) and p. 24 (for SSA).

P. 3, l. 14/15: Is floating ice included in the simulations? If so, what is assumed for the sub-ice-shelf melt rate?

P. 3, l. 16: "till _layer_"?

P. 3, l. 24ff: 30 kyr is likely not long enough to reach thermal equilibrium for an ice body as large as the Greenland ice sheet. This should be commented on. Further, does the inferred sliding depend on the basal thermal state, or is basal sliding applied everywhere?

P. 6, l. 18ff: I cannot see it so well in Fig. 2, but it seems to me that the simulations do not include/reproduce the floating ice tongues (at least off the NEGIS). If so, this may also be partly responsible for the velocity misfits because buttressing effects are missing.

P. 9, l. 7: It would be interesting to quantify this. How large (e.g., in per cent) is the difference between full forcing and (AO+OO)?

P. 10, l. 15: This is the first time in the paper that the enhancement factor is mentioned. It should be defined and specified earlier in the paper (section 2.1).

P. 11, l. 18/19: This should be made a proper reference and cited here as Quiquet and Dumas (2020). And, BTW: _Z_enodo.

––––––––––––––––––––––––––––

---

## Author Comment (AC1) · 20 Oct 2020

*This paper is clearly written and the figures are good. It describes the results of following the ISMIP6 Greenland experimental protocol with a particular dynamical ice-sheet model. Although this is information of use to assessing uncertainties in projections, the scientific gain is not clear. It would be useful if the authors could emphasise scientific lessons we learn from studying this model in particular, beyond its inclusion in the ISMIP6 comparisons, for example? Looking at the conclusions alone, I think a reader who is familiar with the literature of the last several years would find nothing new or surprising, for instance. However, in the paper there are a few new things which ISMIP6 is helping to clarify, and there are moreover useful things which have or could be done with this model, because it is computationally cheap, to test sensitivities.*

It seems to us that such papers that show an individual group contribution to a large intercomparison exercise present three main added values:
- It is a way to document a specific model response for a set of forcings. For example, here, GRISLI shows a sensitivity to climate forcing close to the mean ISMIP6 participating models. This is a potential important information to analyse any further GRISLI results in a broader context.
- The uncertainty that arises from climate evolution (atmospheric and oceanic forcing) can be better quantify in such paper. Although it could also be quantify in the community paper, it is nonetheless only partially address in Goelzer et al. (2020) because of too large material to cover.
- Finally, the ISMIP6 participating models use a wide range of initialisation procedure and they show various biases and model drift. Such issues cannot be discussed in the community paper while it is extensively shown here.
We have added a few information in the introduction section:
"The aim of this paper is to discuss the role of the forcing uncertainties for future projections of the Greenland ice sheet contribution to global sea level rise when using our model. This individual model response can be put in perspective with respect to the multi-model spread discussed in Goelzer et al. (2020). This paper discusses additional experiments not included in the community paper (CMIP6 forcing and separate effects of the oceanic with respect to atmospheric forcing). Compared to Goelzer et al. (2020), we provide here a more detailed description of the initial state and its associated biases and model drift. A companion paper (Quiquet and Dumas, submitted) describes the results for the Antarctic ice sheet."

*A few of my comments relate to the importance of the SMB forcing, which the paper demonstrates. It would be useful to quantify (graphically or in numbers) how much of the spread among GCMs and scenario is due simply to the time-integral of the SMB forcing (as applied to the ice-sheet model), and not affected by the ice-sheet model itself. While it is certainly necessary to use a dynamical ice-sheet model to study large changes in ice-sheets, it would be useful if the authors could present evidence for the need to use one for the 21st century (when not coupled to the atmosphere or ocean), especially as doing so introduces complications of drift and spinup, as described by the paper.*

The spread among GCMs is now shown with a plot of the time evolution of the yearly SMB spatially integrated over the ice sheet.

If we are correct, with the time-integral of the SMB, the reviewer wants to see the SMB contribution to the Greenland melt with respect to the dynamical contribution. However, the time-integral of the SMB as applied to the ice sheet model already accounts indirectly for the dynamical changes because of: i- the SMB correction for the surface elevation change and; ii- the ice mask change. As a result, the time-integral of the SMB will not reflect the impact of SMB only but also, in part, the dynamics. An alternative would be to compute the time-integral of the SMB over a

constant ice sheet topography instead of using the one simulated by GRISLI. Such methods has been widely used in the past (e.g. Fettweis et al., 2013; Meyssignac et al., 2017) as it allows to compute an ice sheet contribution to sea level rise from an atmospheric model only. However this is a crude approximation since the sum will aggregate the strongly negative SMB values at the margin of the ice sheet where the ice will soon disappear and hence overestimate the ice sheet contribution to sea level rise. This overestimation has been quantified with GRISLI to be about 6% (Le clec'h et al., 2019) in 2150 (for 150 simulated years).

We think that the best way to separate the two effect is to compute the dynamical contribution to ice thickness change as explained in Sec. 3.2.3. Note that we also show in this response (Fig. R2) the integrated surface mass balance together with the dynamical contribution to ice thickness change and the ice thickness change, with the same colour scale.

*I have some concern about the prescription of the large melting near the edge (o4 line 12) and the retreat masks (p4 line 32). With both of these enforced, is the dynamical behaviour of the model distorted?*

The very negative SMB outside the present-day ice mask can be seen as a way to correct two type of biases:
- For some areas, the atmospheric forcing computed by MAR presents a positive annual value (ice accumulation) outside the observed present-day ice sheet mask. Uncorrected, this atmospheric forcing bias will result in an overestimation of the ice sheet extent and thickness.
- We use an inverse procedure to infer the basal drag coefficient that best represent the observed ice sheet thickness. By constraining the extent of the ice sheet with an artificial negative SMB, we infer a basal drag coefficient that best reproduce the dynamical behaviour of the ice sheet since the marginal slopes are closer to the observations.
This artificial negative SMB correction does not directly alter the dynamical behaviour of the model but it prevents any ice advance in the future. However, it is probably very marginal for the Greenland ice sheet in the future.

The glacier retreat parametrisation is slightly different. It could alter the dynamics since it is related to an imposed changed in ice thickness. However this is done on purpose, in order to account for a sub-grid process that is not accounted for otherwise. The effect of the glacier retreat can be quantified thanks to the AO experiments. To answer your concern, we have computed the dynamical contribution to ice thickness change (former Fig. 8) for the AO experiments compared to the full forcing experiment. In fact there is only very limited difference between the two (less than 10 metres difference).

*p1 line 10-11. I would not jump to such a strong general conclusion (also on p7).*

Reformulated to:
"Amongst the models tested in ISMIP6, the CMIP6 models produce larger ice sheet retreat than their CMIP5 counterparts."

*p1 line 17-18. I don't think that this statement (of a most likely contribution of 1 m from ice sheets by 2100) is a correct representation of the current state of scientific knowledge. In the first place, you can't state a likelihood independent of scenario, since there are no probabilities for scenarios. Bamber et al. write "For a +5degC temperature scenario, more consistent with unchecked emissions growth, the [median and 95-percentile] are 51 and 178 cm, respectively." I'm not sure what "most likely" means, but 1 m is twice their median. Also, Bamber et al. report an expert elicitation, whose reliability is debatable since it's opaque. For comparison, the AR5 assessment of*

*the likely range of ice-sheet contributions by 2100 under RCP8.5 is 0.09 to 0.28 m from Greenland and -0.08 to 0.14 m from Antarctica.*

We agree with the reviewer. We have chosen to cite the Special Report on the Ocean and Cryosphere in a Changing Climate (SROCC, Oppenheimer et al., 2019) instead of Bamber et al. (2019) here. We have reformulated:
"Amongst the different contributions, the Greenland and Antarctic ice sheets have a potential to raise substantially the global mean sea level, with a weakly constrained trajectory (Oppenheimer et al., 2019)."

*p2 line 1. Why "asymptotic"?*

"approximations" has been replaced by "expansions" since SIA and SSA are the series expansion truncated at the order 0 of the Stokes equation. As such, they are asymptotic expansions.

*p3 line 4. I suppose that strictly you could say an ice-sheet model satisfied momentum conservation, but as far as I know this model and others used for such purposes do not contain terms for acceleration or inertia. That is, momentum is always negligible, and they assume a balance of forces at all times.*

Rephrased to: "[…] that solve the mass conservation and force balance equations".

*p3 Eq 1. I think that BM is a single quantity, isn't it? Typeset like this in a formula it looks exactly like the product of two quantities B and M (like Ubar H is a product). It would be clearer to use a single symbol. Is it just the surface mass balance, or is basal mass balance included too?*

BM has been replaced by M. It is the total mass balance (including basal mass balance). It is now specified in the text.

*p3 line 11. "the total velocity is simply to superposition of the two main approximation". I would suggest "the total velocity is the sum of the velocities predicted in their respective areas by the two main approximations".*

Thanks for your suggestion. We prefer to avoid the use of "respective areas" though, since both velocities are computed for all glaciated grid point. We reformulated as:
"For the whole geographical domain, we assume that the total velocity is the sum of the velocities predicted by the two main approximations: [...]"

*p3 line 15. "for which there is infinite, respectively none, friction at the base." I think this should read "for which there is infinite friction at the base or none, respectively." "None" is a pronoun, not an adjective.*

Thank you, it has been corrected.

*p4 line 5. How accurate are the SMB and the surface topography in the control state?*

The surface mass balance used for the control simulation comes from MAR v3.9. This present-day reference climate is also the one used for the initialisation procedure. This has been clarified:
"This present-day reference climate forcing is used for the initialisation procedure and for the control experiment *ctrl*."
MAR v3.9 is one of the few regional climate models that have been extensively validated against observations. On top of the two reference papers cited, there is an extensive literature that shows the

model performance. We think that MAR offers an accurate representation of the present-day climate over Greenland even though, as any model, it might present some biases (for example a possible overestimation of the precipitation in South-East Greenland, discussed in the manuscript).

Since there is virtually no floating points in the model, the simulated surface topography in the model is the sum of the bedrock topography with the ice thickness. Isostasy being desactivated (now stated in the manuscript), the bedrock topography remains to the one in the observational dataset (Morlighem et al., 2017). Thus, the simulated topography accuracy in the control experiment can be measured by the error on the ice thickness, discussed in Sec. 3.1.

*p4 line 11-14. Does this term strongly interfere with, or even overwhelm, the simulated discharge across the grounding line?*

No, it is only a way to prescribe an ice extent that fits the ice sheet mask in the observations. It has consequences on the initial ice mask and topography and as such it defines the ice dynamics in the initial state (through surface slopes and basal drag coefficient). However, it does not interfere with potential changes in the ice dynamics.

*p4 line 24. State that these are vertical gradients. I would say that they are vertical gradients of surface quantities in the atmosphere model, rather than in the atmosphere.*

Right, we have followed your suggestion:
"[...] yearly values of vertical gradients in the atmospheric model for these two surface variables are also provided."

*p5 line 11. branched to -> branched from.*

Corrected.

*p4 line 21-22. What do you need the surface temperature for, if you're using SMB as forcing?*

Surface temperature is a boundary condition for the temperature diffusion equation. Since the model is thermo-mechanically coupled, temperature affects ice velocities (through viscosity). It will also play a role on the thermal conditions at the base of the ice sheet which also affect ice velocities (frozen grid-points have an infinite friction at the base).

*p5 line 22, p8 line 3, p11 line 11, Fig 5 caption. Although the reader may sympathise with the authors, it's better to avoid "pessimistic" and "optimistic", which are value-judgements.*

Replaced by high emission and low emission scenarios.

*p5 last para. I don't understand the reason for these two experiments. Do they start from the same initial state? Since they have the same forcing, they ought to evolve identically.*

The experiments *ctrl* and *ctrl_proj* have two different initial states since they start at two different dates: 1995 and 2015, respectively. The *ctrl* experiment can be used to quantify the drift in our model during the whole time period (including the historical and the projection). In turn, the advantage of the *ctrl_proj* experiment is to be directly comparable to the projection experiments as they cover the same time period and they use the same initial state (which was not the case with the *ctrl* experiment). To clarify this point, we added the following:

"The *ctrl* experiment can be used to quantify the simulated model drift over the whole time period (1995-2100). Instead, the *ctrl_proj* can be directly used to quantify the importance of climate forcing evolution since it uses the same initial state in 2015 as the different projection experiments."

*p5 line 34. alike -> like.*

Corrected.

*p6 line 22. best -> better.*

Corrected.

*p6 line 24. "In doing so" means doing what? - absolute or logarithm? I would have assumed logarithm, but the next sentence suggests otherwise. What are the units of 0.55? What are the units of velocity before taking the logarithm? (Strictly you can only take the log of a dimensionless quantity, but the conversion factor between different velocity scales will be a constant offset in the log so doesn't affect its RMSE, I suppose.)*

We meant logarithm of the velocity (expressed in metre per year but as you rightly point out an other choice will not affect the RMSE). We have rephrased to:
"When using the logarithm of the velocity GRISLI slightly improves compared to the other participating models since the RMSE is about 0.55 (eleventh worst value out of 21). This means that the error [...]"

*p6 line 30. Why is this "on the contrary"? If I read this correctly, all the errors are in the same direction (too slow in the model). Can you suggest the reason for this systematic bias? What implication does it have for projections?*

It should have been "On the contrary, the South East glaciers, Kangerdlugssuaq and Helheim, are too fast in the model." (and not "too slow"). There is no systematic biases for the velocity: amongst the largest ice streams, the NEGIS, Petermann and Jakobshavn are too slow but the Kangerdlugssuaq and Helheim are too fast.

*p7 line 4-5. What implication will this bias in SMB have for projections?*

It is difficult to give a definite answer to this question. An overestimation of the precipitation might moderate the effect of the expected decrease in SMB in the future. However a too wet climate could also be the sign of a too intense penetration of warm (thus humid) air over this area which could also facilitate melting at high elevation. Such atmospheric processes are best quantified with dedicated atmospheric model experiments.

*p7 line 9 and 15. Are these large drifts in thickness and velocity related? What effect will they have on projections? It's not obvious that you can simply subtract an unforced drift when it's large compared with the forced response.*

The drift in thickness and velocities are related since the two variables are tightly coupled together in the model. However, we think that the velocity drift mostly derived from the ice thickness drift. For example, the ice thickness drift in South-East Greenland near the Helheim glacier is negative, which induces a decrease of the ice velocity (less ice to export).

In the paper, the plots of the time evolution of integrated variables show the control experiments (i.e. the drift) as well as the projections without the drift subtraction. We subtract the drift only for

2D maps to better highlight the impact of climate change. However, the drift shown in Fig. 1 (original manuscript) is small when compared to the ice thickness change induced by climate change.

*p7 line 18. start by -> start with.*

Corrected.

*p7 line 21-24. Presumably this spread comes mostly from the spread in SMB forcing from the GCMs. Could you also add the ice-sheet area- and time-integral of the SMB perturbation to the graphs?*

In addition to the response we made earlier on your main comment, we can add a few information here. We have preferred to not plot the time integral of the mean SMB over the ice sheet since it may be more difficult to interpret than the yearly evolution. The time integral of this variable is essentially positive with only negative values for some models towards the end of the century. This is because the area-integrated SMB becomes negative only after 2060 for some models (and remains positive for others). Since the simulated ice sheet shows only a small drift in the control experiment, the positive area-integrated at the beginning of the simulation is almost balanced by the melt at the base of the ice sheet and the calving flux. Thus, the time integral of the spatial mean SMB draws an incomplete picture of the evolution of ice volume and does not allow for a separation of the ice dynamics versus SMB contribution.
Nonetheless, to show the spread amongst GCMs, we have added the time evolution of the SMB integrated over the ice sheet mask and added a few sentences:
"The differences in ice volume evolution are tightly linked to the surface mass balance evolution for the different climate forcing. Amongst the CMIP5 climate models, IPSL-CM5-MR and MIROC5 simulate a mean surface mass balance negative as early as 2060 while it remains positive over the next century for CSIRO-Mk3.6 (Fig. 4)."

*p7 line 21-24. It seems that these projections imply quite a low sensitivity to climate change compared with the models on which the AR5 was based; their assessment of the Greenland contribution by 2100 under RCP8.5 is 90-280 mm, of which 40-220 mm is from SMB change.*

Although slightly smaller perhaps, GRISLI shows a climate sensitivity close to the mean of the ISMIP6 participating model. The community paper (Goelzer et al., 2020) reports a range of 70-135 mm (mean of 100 mm) using MIROC5 RCP8.5 while GRISLI shows a range of 75-95 mm (low to high oceanic sensitivity) under the same forcing. This is now specified in the text:
"The 2100 sea level contribution simulated by GRISLI is close to the mean model response amongst the ISMIP6 participating models."
The numbers in the AR5 for RCP8.5 were larger (Table 13.5 reports 0.07 to 0.21 m from which 0.03 to 0.16 m due to SMB change). However, these estimates were derived only from a small number of studies/models, compared to the 21 ice sheet models in ISMIP6. They were also obtained sometimes with a simpler methodology : the Special Report on the Ocean and the Cryosphere in a Changing Climate (SROCC) reports a median value for process-based approaches of 11.9 cm under RCP8.5.

*p7 line 25. What sort of "tipping point" do you have in mind, that you might see in the volume evolution? Can you give references to relevant suggestions?*

We were imprecisely referring to a sharp change of slope. This has been reformulated:
"However, we can not distinguish any sharp inflexion in the volume evolution over the next century."

*p7 line 26-27. I think we should be more cautious in drawing conclusions. There are only four CMIP6 models considered in this study, out of dozens in total, and two of the four are at the edge of the CMIP5 distribution in your projections. Only two show much greater sensitivity, and those results are within the AR5 range.*

Thanks for pointing this out. We have reformulated:
"CMIP6 models show generally a much larger Earth climate sensitivity than their equivalent in the former CMIP5 generation (Forster et al., 2020). In particular, the CMIP6 models used in ISMIP6 have an Earth climate sensitivity from 4.8 to 5.3, i.e. larger than the CMIP5 models used here, which show a range from 2.7 to 4.6 (Meehl et al., 2020)."

*p8 line 6-7. It's not the GHG itself which is the driver, but the warming it produces; that is also the reason why the rate of mass loss goes up with time, and the main reason for the spread among models.*

Reformulated:
"The future atmospheric and oceanic warming induced by the greenhouse gas mixing ratio is thus a major driver for the Greenland ice mass at the century time scale."

*p8 line 13-14. Since the point you wish to make is the similarity of the patterns, it would be better to show these maps divided by the integrated change in each case i.e. normalised to the same GMSLR contribution. That would reveal the patterns themselves, so they could be compared, which I agree should be the purpose of this figure.*

Such figure is shown below in this response (Fig. R1). It is true that the new figure shows nicely the similarity of the patterns for the different GCMs. However, we think that the absolute ice thickness change for a given climate forcing is more informative for the reader as it is a way to show how the volume change (integrated value) translates into ice thickness change. However, if the reviewer believes that we should add this figure in the supplementary material, we would be happy to do so.

[Figure]

**Figure R1.** Simulated ice thickness change (2100 – 2015) normalised its spatial average (i.e. volume change) for: (a) CSIRO-Mk3.6 (RCP8.5); (b) MIROC5 (RCP8.5); (c) MIROC5 (RCP2.6) and; (d) UKESM-CM6 (SSP585) climate forcing. The medium oceanic sensitivity has been used for this figure, except for UKESM-CM6 (d) for which we use the high oceanic sensitivity.

*p9 line 6. It would be interesting to see the time-integral of the applied SMB perturbation here, to compare with the AO experiments (as I also suggested on p7 for Fig 3). Any difference is due to the dynamical response to the SMB forcing.*

The integrated SMB indirectly accounts for dynamical changes. First through the elevation feedback on SMB with the vertical lapserate. Second because the ice mask can change due to ice dynamics. will reflect indirectly the dynamical response, through the elevation change correction and ice mask change. We do not think that such a figure will allow to distinguish the dynamical response from the SMB forcing.

In Fig. R2 of this response, we show the integrated surface mass balance together with the dynamical contribution to ice thickness change and the ice thickness change, with the same colour scale.

*p9 lines 16-23. The text says "Fig. 8b shows the difference in ice flux convergence in 2100", and the fig caption says "change in the dynamic contribution to ice thickness change in 2100". I don't think either of those is a correct description, if I have understood correctly. You also say, "This can be considered as the dynamical contribution to ice thickness change," which I think is correct. The quantity shown is the difference (change in topography during the experiment) minus (time-integral during the experiment of the local mass balance change with respect to control) - is that right? It would be useful to compare this difference with the change in topography in the same experiment, using the same color scale, in order to see the relative importance of the dynamical change. If it's a small fraction, you might argue that there's no need to use a dynamical ice-sheet model for projections on this timescale. Where it's not small, you can comment. Part of the dynamical contribution near the coast is a response to the ocean forcing, I presume. Therefore it would also be useful to show the same comparison for the AO experiment. That is, would it be good enough to make the projection without a dynamical model, simply by time-integrating the local SMB perturbation?*

Yes you are right with the definition and thank you for pointing this terminology inconsistencies. It is now referred as "dynamical contribution to ice thickness change" throughout the manuscript.

We have added the ice thickness difference in Fig 9, to compare with the dynamical contribution to ice thickness change and added a few information in this manuscript:
"The integration in time of Eq. 1 over 2015-2100 suggests that the integrated ice flux convergence is the difference between the ice thickness change from 2015 to 2100 and the integrated mass balance (surface and basal mass balance and calving) over this period. The integrated ice flux convergence can be considered as the dynamical contribution to ice thickness change. It should be noted that the integrated mass balance here also includes the effect of ice mask change and surface elevation change. As such, it is not comparable to what would have been obtained with an atmospheric model only. Fig. 9b shows the difference of the dynamical contribution in 2100 for a selected climate forcing with respect to the control ctrl_proj experiment. The pattern mostly follows the one of velocity change (Fig. 9a). There is an important positive dynamical contribution to ice thickness change (ice flux convergence) at the margins that tends to partially compensate the decrease in surface mass balance. Conversely, upstream regions show a slightly negative dynamical contribution (ice flux divergence). This pattern is similar amongst the different climate forcings. To compare the relative importance of the dynamical contribution with respect to surface mass balance to explain the ice thickness change we show the ice thickness change in 2100 with the same colour scale in Fig. 9c. The dynamical contribution shows generally much smaller value suggesting that surface mass balance explains the largest changes in ice thickness. However, locally, for example in the South-East and central West regions the dynamical contribution can be the largest driver of ice thickness change."

Fig. R2 is the same as Fig. 9 in the paper, the only difference is that it shows the integrated surface mass balance as well. The dynamical contribution is directly constructed from the difference of the ice thickness change and the integrated total mass balance (from which surface mass balance is the main driver). In the paper, we keep the version of the figure with the dynamical contribution to ice thickness change together with the ice thickness change, but we omit the integrated surface mass balance since we do not think it brings an additional value.

There is virtually no change in the dynamical contribution to ice thickness change when comparing the standard experiment to the AO experiment. The glacier retreat parametrisation can be seen as a calving process. It implies a slightly greater ice thickness change but its effect is affected to the integrated mass balance change (which include surface and basal mass balance in addition to calving). The difference in thickness and surface slope change between the AO and standard experiment does not seem to be sufficiently large to affect the ice dynamics.

[Figure]

**Figure R2.** Simulated surface velocity change during the projection run (2096-2100 with respect to 2015-2019) using MIROC5 forcing under RCP8.5 with a medium oceanic sensitivity. **b**: change in the dynamical contribution to ice thickness change in 2100 (see text for definition) for this same experiment. **c**: simulated ice thickness change (2100-2015). **d**: time integral of the surface mass balance (2015-2100). For all panels, we corrected the changes by the ones simulated in the control experiment *ctrl_proj* over the same period. Note that the colour scale is not symmetrical for **(b)**, **(c)** and **(d)**.

*p9 line 30. As a guide to the possible magnitude of this underestimate, you could state what the presently observed ice-sheet imbalance would give if it continued as a constant rate to 2100 and compare with your projected changes in response to forcing.*

This is a very interesting point indeed, and maybe one of the major point of this paper but also the community paper. Our ice sheet models do not reproduce the recent accelerations and as such probably bias our projections towards low estimates. We have added the following:
"This means that, by constructions, our simulations underestimate the Greenland ice sheet contribution to future sea level rise. A simple linear extrapolation of the 2006-2016 rate (0.77 mm yr$^{-1}$, Oppenheimer et al., 2019) up to 2100 would result in a 6.5 cmSLE from the Greenland ice sheet. This number is large compared to the GRISLI results discussed in this paper, and more generally it is large compared to the spread amongst ISMIP6 models (3.5 to 14 cmSLE, Goelzer et al., 2020). This suggests that model initialisation is one of the largest source of uncertainty for

model projections. Instead of using a methodology that produces ice sheet at equilibrium, some promising alternatives exist, [...]"

*p10 line 4-16. This is useful, but it's not really discussion, I'd say. It's another sensitivity test, and it would go well in sect 3.2.3 about change in ice dynamics.*

We have moved this part in the Sec. 3.2.3.

*p10 line 20. Why is it necessarily an overestimate?*

Because the diffusion of the cold temperature within the ice sheet is not accounted for. This is now clarified:
"Our internal temperature field is the result of a long thermo-mechanical equilibrium under perpetual present-day forcing and as such, it is necessarily overestimated since the diffusion in the ice sheet of the cold temperature of the glacial period is not accounted for."

*p10 line 27-29. Yes, it would! Since your model is particularly computationally inexpensive, please could you do it and tell us the answer? :-)*

Since we think that it makes little sense to perform long multi-millenial integrations with a constant prescribed basal drag coefficient, we are currently working on the calibration of the model parameters for an interactive computation of the basal drag coefficient as in Quiquet et al. 2018. However, although our model is relatively cheap it nonetheless currently requires 11 days on our local computers to perform 10 kyr with the 5-km grid resolution used in the paper. Hopefully in the future we will be able to show the behaviour of our model for two completely independent initialisation procedure.

*p10 line 31-32. Could you quantify the elevation-SMB feedback here, or earlier, and compare it with Edwards et al. (Cryosphere, 2014)? You could directly quantify it by running a sensitivity test in which the lapse-rate adjustment is excluded, I suppose.*

We have performed a sensitivity experiment using MIROC5 RCP8.5 and the medium oceanic sensitivity in which we did not account for the lapse-rate correction. We found a reduction by 5.1% of the Greenland contribution to sea level rise in this experiment with respect to its counterpart in which the correction is applied. This number is close to the 4.3 reported by Edwards et al. (2014). We have added the following:
"The forcing methodology used for ISMIP6-Greenland does account for the vertical elevation feedback on temperature and surface mass balance. In order to quantify the impact of this correction on the simulated evolution of the ice sheet, we run a sensitivity experiment in which this correction is not accounted for. Using MIROC5 under RCP8.5 scenario with a medium oceanic sensitivity, we simulate a Greenland contribution to future sea level rise 5.1% smaller in this sensitivity experiment compared to the same experiment in which the vertical correction is applied. This number is slightly higher than the effect reported by Edwards et al. (2014) and Le clec'h et al. (2019a) (4.3 and 4.2% respectively) but smaller to Vizcaino et al. (2015) (8-11%) and Calov et al. (2018) (about 13%). Differences in resolution and/or physical processes implemented in the atmospheric model could explain this diversity."

*p11 line 8. is systematically loosing -> systematically loses*

Corrected.

*Fig 1 caption. Does "respective to" mean "with respect to"? For clarify please state the years of the end of the historical and end of ctrl_proj.*

Done.

**References**

Calov, R., Beyer, S., Greve, R., Beckmann, J., Willeit, M., Kleiner, T., Rückamp, M., Humbert, A., and Ganopolski, A.: Simulation of the future sea level contribution of Greenland with a new glacial system model, The Cryosphere, 12, 3097–3121, doi:https://doi.org/10.5194/tc- 12-3097-2018, https://www.the-cryosphere.net/12/3097/2018/, 2018.

Edwards, T. L., Fettweis, X., Gagliardini, O., Gillet-Chaulet, F., Goelzer, H., Gregory, J. M., Hoffman, M., Huybrechts, P., Payne, A. J., Perego, M., Price, S., Quiquet, A., and Ritz, C.: Effect of uncertainty in surface mass balance–elevation feedback on pro- jections of the future sea level contribution of the Greenland ice sheet, The Cryosphere, 8, 195–208, doi:10.5194/tc-8-195-2014, http://www.the-cryosphere.net/8/195/2014/, 2014.

Fettweis, X., Franco, B., Tedesco, M., van Angelen, J. H., Lenaerts, J. T. M., van den Broeke, M. R., and Gallée, H.: Estimating the Greenland ice sheet surface mass balance contribution to future sea level rise using the regional atmospheric climate model MAR, The Cryosphere, 7, 469–489, https://doi.org/10.5194/tc-7-469-2013, 2013.

Goelzer, H., Nowicki, S., Payne, A., Larour, E., Seroussi, H., Lipscomb, W. H., Gregory, J., Abe-Ouchi, A., Shepherd, A., Simon, E., Agosta, C., Alexander, P., Aschwanden, A., Barthel, A., Calov, R., Chambers, C., Choi, Y., Cuzzone, J., Dumas, C., Edwards, T., Felikson, D., Fettweis, X., Golledge, N. R., Greve, R., Humbert, A., Huybrechts, P., Le clec'h, S., Lee, V., Leguy, G., Little, C., Lowry, D. P., Morlighem, M., Nias, I., Quiquet, A., Rückamp, M., Schlegel, N.-J., Slater, D. A., Smith, R. S., Straneo, F., Tarasov, L., van de Wal, R., and van den Broeke, M.: The future sea-level contribution of the Greenland ice sheet: a multi-model ensemble study of ISMIP6, The Cryosphere, 14, 3071–3096, doi:https://doi.org/10.5194/tc-14-3071-2020, 2020.

Le clec'h, S., Charbit, S., Quiquet, A., Fettweis, X., Dumas, C., Kageyama, M., Wyard, C., and Ritz, C.: Assessment of the Greenland ice sheet–atmosphere feedbacks for the next century with a regional atmospheric model coupled to an ice sheet model, The Cryosphere, 13, 373–395, doi:10.5194/tc-13-373-2019, 2019.

Meyssignac, B., Fettweis, X., Chevrier, R., and Spada, G.: Regional Sea Level Changes for the Twentieth and the Twenty-First Centuries Induced by the Regional Variability in Greenland Ice Sheet Surface Mass Loss, J. Clim., 30, 2011–2028, https://doi.org/10.1175/JCLI-D-16-0337.1, 2017.

Oppenheimer, M., Glavovic, B. C., Hinkel, J., van De Wal, R. S. W., Magnan, A. K., Abd-Elgawad, A., Cai, R., CifuentesJara, M., DeConto, R. M., Ghosh, T., Hay, J., Isla, F., Marzeion, B., Meyssignac, B., and Sebesvari, Z.: Sea Level Rise and Implications for Low-Lying Islands, Coasts and Communities, in: IPCC Special Report on the Ocean and Cryosphere in a Changing Climate, edited by: H.-O. Pörtner, D. C. R., V. Masson-Delmotte, P. Zhai, M. Tignor, E. Poloczanska, K. Mintenbeck, A. Alegría, M. Nicolai, A. Okem, J. Petzold, B. Rama, N. M. Weyer, 2019.

Quiquet, A., Dumas, C., Ritz, C., Peyaud, V., and Roche, D. M.: The GRISLI ice sheet model (version 2.0): calibration and validation for multi-millennial changes of the Antarctic ice sheet, Geoscientific Model Development, 11, 5003–5025, doi:10.5194/gmd-11-5003-2018, 2018.

Vizcaino, M., Mikolajewicz, U., Ziemen, F., Rodehacke, C. B., Greve, R., and van den Broeke, M. R.: Coupled simulations of Greenland Ice Sheet and climate change up to A.D. 2300, Geophysical Research Letters, 42, 2014GL061 142, doi:10.1002/2014GL061142, 2015.

---

## Author Comment (AC2) · 20 Oct 2020

*In this manuscript, the authors report on their ISMIP6 Greenland projections with the model GRISLI. The paper is easy and straightforward to follow. Its scientific value beyond the community publication (Goelzer et al., 2020, in press) lies in a more detailed description of the set-up of GRISLI, a more detailed analysis of the results and the fact that the entire suite of ISMIP6 experiments (Tier 1-3) are dealt with.*

*Overall, I found the results interesting and the presentation adequate. I'd only like to raise some issues that should be dealt with as follows:*

Thank you for your positive evaluation, we reply to your individual comments below.

*The English writing clearly has some room for improvements. I am not going to point out all the issues, but just some examples from the first page: P. 1, l. 3/4: "an increase_d_ mass loss". P. 1, l. 5: "the largest single source contribution _after_ the thermosteric contribution". P. 1, l. 19/20: Assessment of projections? Either "need for assessment of future SLR by projections" or "need for projections of future SLR". P. 1, l. 22: Strange formulation: "from changing boundary conditions such as climate change". Before resubmission, the entire manuscript should be very carefully proof-read by a (near-) native speaker or a professional language editing service.*

Thank you for your corrections, we have followed all your suggestions. We are indeed non-native English speakers but we put a lot of effort to write in English since it is the international language for Science. Even after careful reading, we are aware that our manuscript will contain grammatical errors or poorly formulated sentences but we think that it is generally understandable. If not, we will be more than happy to follow your corrections. Also, it might be relevant to note that The Cryosphere journal includes a language editing service for all accepted manuscripts.

*Throughout MS (e.g., p. 1, l. 10, l. 14): "mmSLE" -> "mm SLE"*

Corrected.

*Throughout MS (e.g., p. 2, l. 4): "in term of" -> "in terms of"*

Corrected.

*P. 1, l. 17/18, "most-likely amplitude exceeding 1 metre in 2100": This is not what has been found in the ISMIP6 ensemble projections (Goelzer et al., 2020, TC, in press; Seroussi et al., 2020, TC, in press). Even with the most sensitive model results, it is less than half a metre combined. At some point in the paper, this should be mentioned.*

We apologise, this was an exaggerated statement since the range of Bamber et al. (2019) is 51 to 178 cm for unmitigated emissions. We have chosen to cite the Special Report on the Ocean and Cryosphere in a Changing Climate (SROCC, Oppenheimer et al., 2019) instead of the expert judgement of Bamber et al. (2019) here:
"Amongst the different contributions, the Greenland and Antarctic ice sheets have a potential to raise substantially the global mean sea level, with a weakly constrained trajectory (Oppenheimer et al., 2019)."

The numbers for the contribution of the Greenland ice sheet to 2100 sea level rise in Goelzer et al. (2020) and in our manuscript are within the range of the SROCC.

*P. 3, l. 7, 17: Add commas after the displayed equations.*

Done.

*P. 3, l. 11-13: The description of the SIA and SSA is over-simplified. Starting from full Stokes, in both cases, some horizontal and some vertical derivatives of the components of the stress tensor are neglected. In very compact form, this is shown in the tutorial at http://doi.org/10.5281/zenodo.3739009, p. 22 (for SIA) and p. 24 (for SSA).*

Reformulated:
"For the whole geographical domain, we assume that the total velocity is the sum of the velocities predicted by the two main approximations: the shallow ice approximation (SIA) in which the deformation is entirely driven by the vertical shear and the shallow shelf approximation (SSA) in which the vertical shear is neglected and the horizontal stresses are predominant."

*P. 3, l. 14/15: Is floating ice included in the simulations? If so, what is assumed for the sub-ice-shelf melt rate?*

Only few glaciers in Greenland present a floating tongue and when they do it is located in very narrow valleys. The 5-km grid used in our model is not precise enough to represent these floating tongues and the physical processes related. This is why we have imposed a very large basal melting rate in our simulations (200 m yr$^{-1}$) to avoid floating points. This is now stated in the manuscript:
"Since 5 km is too coarse to represent Greenland floating ice tongues, sub-shelf melting rate has been set to a large value (200 m yr$^{-1}$) to discard simulated floating points."
We agree that is a simplification that can bias the future projections. In fact, this is not a problem specific to GRISLI since most ISMIP6 participating models do not have the resolution needed to represent such floating ice tongues. The ISMIP6 glacier retreat parametrisation has been developed (Slater et al., 2019) to account for such a process in models that would not otherwise.

*P. 3, l. 16: "till _layer_"?*

Corrected.

*P. 3, l. 24ff: 30 kyr is likely not long enough to reach thermal equilibrium for an ice body as large as the Greenland ice sheet. This should be commented on. Further, does the inferred sliding depend on the basal thermal state, or is basal sliding applied everywhere?*

We agree. This is why we performed more than ten cycles (thermal equilibrium + multiple 200-yr simulations). The basal drag coefficient and the basal temperature are coupled and in doing more cycles, we expect the two variables to be consistent with each other. This is now more clearly stated in the manuscript:
"After a few 200-yr experiments, we repeat the thermal equilibrium computation restarting from the previous equilibrium state with the newly inferred basal drag coefficient. In doing so, the basal drag coefficient and the temperature at the base are consistent with each other."

*P. 6, l. 18ff: I cannot see it so well in Fig. 2, but it seems to me that the simulations do not include/reproduce the floating ice tongues (at least off the NEGIS). If so, this may also be partly responsible for the velocity misfits because buttressing effects are missing.*

You are correct, we do not simulate the floating ice tongues which can exert buttressing. However, the simulated velocity in the NEGIS is underestimated so it cannot be linked to the missing

buttressing (which would reduce the velocity even more). The velocity misfit is most probably linked to the basal drag coefficient.

*P. 9, l. 7: It would be interesting to quantify this. How large (e.g., in per cent) is the difference between full forcing and (AO+OO)?*

The AO+OO explains 93.6%, 91.6% and 92% of the full forcing for MIROC5, NorESM1 and CSIRO-Mk3.6 respectively. This is now stated in the manuscript:
"Also, the sum of the ice loss of AO and OO experiments approximate closely the ice loss simulated when using the full forcing (92 to 94% of the full forcing). "

*P. 10, l. 15: This is the first time in the paper that the enhancement factor is mentioned. It should be defined and specified earlier in the paper (section 2.1).*

Added in the description of the model:
"Like most ice sheet model, GRISLI uses a flow enhancement factor that favours longitudinal deformation in the SIA (Quiquet et al., 2018). However, here we use a flow enhancement factor set to 1 (no enhancement). Similarly, the flow enhancement factor for the SSA is also set to 1."

*P. 11, l. 18/19: This should be made a proper reference and cited here as Quiquet and Dumas (2020). And, BTW: _Z_enodo.*

Done.

---

## Author Response (AR2)

We would like to thank again the reviewers for their helpful comments. We reply to each individual comment in the following.

**Anonymous Referee #1**

*I thank the authors for their thoughtful consideration of the comments and the improvements they propose to make to the paper. I appreciate the perspective they have added in the introduction, and the clarification of the decomposition into SMB and dynamic contributions. I have a few more comments on their new draft which they may wish to consider.*

*page 1 line 16-17. "and is now larger than"*

Done.

*page 4 line 6. If I've understood this correctly, you call them "iterations" because you do them sequentially. "Iteration" usually has the idea of repeating the procedure but starting from where the previous iteration ended, whereas your "iterations" all start from the same state. Therefore I find "iteration" slightly misleading. You could call them an ensemble, except that you also use the final state to modify the next "iteration". Perhaps it would be better to call them something like "short transient experiments", which distinguishes them from the long equilibrium experiments.*

The procedure intends to infer the basal drag coefficient that leads to the best match with observed present-day ice thickness. After each short transient experiments we update the basal drag coefficient, starting from an initial guess. With these updates, we are converging towards the minimal ice thickness error with respect to the observations (reachable to our model given the forcings and initial state). It seems that this is in line with your definition of an iterative procedure (the basal drag coefficient is updated at each iteration). What might be confusing perhaps is that the thermal state and the geometry are not changed from one iteration to an other. This is because they are not the variables that we intend to infer during this step.

The procedure also includes a simple way to compute the initial thermal state as well (long simulation with fixed geometry). Here again, it is iterative since from one long iteration to an other the temperature is updated.

We slightly rephrase a sentence:
"Each 200-yr iteration uses the exact same initial condition for the ice thickness and temperature but have a different initial basal drag coefficient."

*page 6 line 16. "Instead" doesn't sound quite right to me. You don't mean a replacement, but a comparison. I would say, "On the other hand" or "By contrast".*

We now use "By contrast", thanks for the suggestion.

*page 8 line 19. I don't think it's correct that "CMIP6 models show generally a much larger" climate sensitivity than CMIP5. There are certainly some models which a considerable larger sensitivity, but the ranges given on the following lines for ISMIP6 are not representative of CMIP6 in general.*

We agree that the subset used for ISMIP6 is not necessarily representative for the whole ensemble.

We simplified to:

"The CMIP6 models used in ISMIP6-Greenland have an Earth climate sensitivity from 4.8 to 5.3 °K, i.e. larger than the CMIP5 models used here, which show a range from 2.7 to 4.6 °K (Meehl et al., 2020)."

*page 8 lines 20-21. Units of climate sensitivity are K.*

Corrected, thanks for noticing.

*page 10 line 27. I think you mean it's the same range of colors, don't you? I would not say it's the same color scale; it's an order of magnitude different, which is the point you are making, if I understand correctly.*

In fact the colour scale is mirrored, i.e. opposite values and invert colour gradient, since the dynamical contribution is generally negative (resp. positive) where the ice thickness change is positive (resp. negative). For positive values, the dynamical contribution is one order of magnitude larger than the ice thickness change, when it is one one order of magnitude lower for negative values. We corrected the sentence:
"[…] we show the ice thickness change in 2100 with a similar colour scale (opposite values and invert colour gradient) in Fig. 10c.

*page 11 line 9. I would say "inaccurate" rather than "unjustified", which is unfair to yourselves. It was a justifiable choice, on grounds of simplicity.*

Thanks, corrected.

*Fig 11. It would be helpful if you could put a % change on the right-hand axis of these figures.*

We are not sure of what you want us to do. It seems rather unusual to put the unit of an axis variable at the end of this axis. The units (here %) are specified with the axis label for all the plots.

*page 12 line 2-15. This point seems important to me and perhaps deserves to be mentioned in the conclusions and maybe the abstract.*

The last sentence of our conclusion now reads:
"Finally, the initial condition chosen for the ice sheet model remains an important topic for ice sheet modelling. In particular, assuming an ice sheet in equilibrium with present-day climate for the initial condition, as done here but also in most ISMIP6 participating model, could lead to an underestimation of the future mass loss."

*page 12 line 5. loosing -> losing (you can't be blamed for surprising English spellings)*

Thanks, corrected.

*page 13 line 18. looses -> loses (same comment)*

Corrected.

*page 13 line 17-27. As I commented last time, I think you could draw attention in the conclusions to some more of the findings from the large range of experiments you are able to carry out (as well as adding the point of the first paragraph of page 12). You could state more clearly that the dynamical contribution is generally much smaller (by an order of magnitude) than the SMB contribution (which is implied by the final sentence of the conclusions, but clearer earlier in the text), that*

*consequently the uncertainty in ocean forcing has a relatively small effect on the spread of projections, which mostly comes from the SMB, and that the basal drag is rather poorly constrained but doesn't much affect the projections to 2100. Perhaps some of these might be added in the abstract too.*

We have followed your suggestions and expanded the conclusion section with the following:
"The oceanic forcing contributes to ice loss by about 10~mm~SLE in 2100. In addition, the time integral of the surface mass balance is generally much larger than the dynamical contribution to ice thickness change (by an order of magnitude). This suggests that the Greenland ice mass loss in the future is mostly driven by surface mass balance changes, in particular through a larger ablation at the ice sheet margin. This process should thus be carefully implemented in ice sheet models aiming at simulating the Greenland ice sheet evolution at the century scale. With additional sensitivity experiments, not included in ISMIP6, we have also shown that the choice of uncertain mechanical parameters (i.e. flow enhancement factor and basal drag coefficient) has only a small impact on the spread of mass loss. Finally, the initial condition chosen for the ice sheet model remains an important question for ice sheet modelling. In particular, assuming an ice sheet in equilibrium with present-day climate for the initial condition, as done here and in most ISMIP6 participating models, could lead to an underestimation of the future mass loss. "

In addition, the end of the abstract now reads:
"The oceanic forcing contributes to about 10~mm~SLE in 2100 in our simulations. In addition, the dynamical contribution to ice thickness change is small compared to the impact of surface mass balance. This suggests that mass loss is mostly driven by atmospheric warming and associated ablation at the ice sheet margin. With additional sensitivity experiments we also show that the spread in mass loss is only weakly affected by the choice of the ice sheet model mechanical parameters."

**Anonymous Referee #2**

*Overall, I think that the authors have done a reasonably good job revising the paper, and I only have some remaining issues.*

*Most importantly, I take it from the authors' reply that they have not followed my suggestion that "the entire manuscript should be very carefully proofread by a (near-) native speaker or a professional language editing service." As a result, there are still a considerable number of language issues, especially in the newly written parts. I am aware that, as the authors write, "The Cryosphere journal includes a language editing service for all accepted manuscripts". However, in my understanding, this service must not serve as an excuse for taking a relaxed stance on the quality of the writing. Just being "generally understandable" is not sufficient! Rather, authors of papers in high-quality journals like The Cryosphere should make every effort to deliver their contributions in a near-perfect condition.*

We understand. However, we do not have the fundings in our projects to use a professional language editing service for our papers. Some native speaker colleagues offer some help occasionally, but it cannot be on a regular basis since it is time consuming. As such we are somehow limited by our English skills.

*Below, I will suggest some corrections (the majority of all comments). However, I neither have the time nor the motivation to proofread the entire manuscript carefully (especially because I'm a non-native speaker either...), so that these corrections have no claim for completeness.*

Thank you very much for your valuable suggestions.

*Page/line numbers refer to the diff version of the manuscript that came attached to the response letter.*

*P. 2, l. 28:*
*"(CMIP6 forcing, separate effects of the atmospheric and oceanic forcings)"*

Done.

*P. 3, l. 26: "ice sheet model_s_"*

Corrected.

*P. 3, l. 26:*
*This sounds as if the flow enhancement factor favours only selected deformation modes., which is wrong. Rather "factor that increases the ice fluidity in the SIA".*

Thanks for the suggestion.

*P. 4, l. 9: "consists _of_ finding"*

Corrected.

*P. 4, l. 27: "Greenland_'s_ floating ice tongues, _the_ sub-shelf melting rate"*

Corrected.

*P. 5, l. 5: Delete "in the atmospheric model"*

Done.

*P. 7, l. 12: Unit of "about 0.55"? Log of velocities in m yr-1, I suppose?*

Correct. Precision added.

*P. 7, l. 18: "the _south-eastern_ glaciers"*

Corrected.

*P. 7, l. 19: "small_,_ meaning"*

Corrected.

*P. 8, l. 14:*
*"sharp inflexion" sounds strange either. What about "we can not discern any sudden change ... century that may indicate a tipping point".*

Again, thank you for your suggestion.

*P. 8, l. 16: "forcing_s_"*

Corrected.

*P. 8, l. 16: "mass balance _becoming_ negative"*

Added.

*P. 8, l. 17: "2060_,_ while"*

Added.

*P. 8, l. 23: "_the_ SSP585 scenario"*

Corrected.

*P. 8, l. 26:*
*The authors may consider comparing their findings re CMIP5 vs. CMIP6 to those described in Sect. 4.2 of the Technical Report by Greve et al. (2020, Zenodo, https://doi.org/10.5281/zenodo.3971251).*

Thanks for the suggestion. We have added:
This has also been reported by Greve et al. (2020) where the use of the CMIP6 model ensemble under the SSP585 leads to an ice sheet contribution to sea level rise increased by at least 70 % with respect to the contribution simulated using the CMIP5 ensemble.

*P. 8, l. 32: "mass loss _compared_ to the"*

Corrected.

*P. 9, l. 1: "_compared_ to the"*

Corrected.

*P. 10, l. 3: "of the _combined_ forcing"*

Corrected.

*P. 10, l. 15: "integrated _divergence of the ice flux_ can be"*

Corrected.

*P. 10, l. 21: Delete "(ice flux convergence)" [unnecessary duplication]*

Done.

*P. 10, l. 22: Delete "(ice flux convergence)"*

Done.

*P. 10, l. 26: "change_,_ we show"*

Added.

*P. 10, l. 27: "values_,_ suggesting"*

Added.

*P. 11, l. 5: "south-eastern and central western regions"*

Corrected.

*P. 11, l. 27: "_for_ the control experiment"*

Corrected.

*P. 12, l. 3: "optimally tune_s_"*

Corrected.

*P. 12, l. 4: "with _the_ present-day"*

Corrected.

*P. 12, l. 6: "by construction, our simulations"*

Corrected.

*P. 12, l. 7-9: 6.5 cm is not "large" compared to 3.5-14 cm. Rather "it is comparable to the spread"?*

You are right. This sentence now reads:
"This number is comparable to the GRISLI spread discussed in this paper, and more generally to the spread amongst ISMIP6 models (3.5 to 14 cm SLE, Goelzer et al., 2020)"

*P. 12, l. 10: "largest source_s_ of uncertainty"*

Corrected.

*P. 12, l. 11: "_an_ ice sheet at equilibrium"*

Corrected.

*P. 12, l. 19: Not only diffusion. Except for the near-basal parts of an ice sheet, downward advection is actually more efficient. Perhaps more generally "since the information|memory of the low temperatures of the glacial period in the ice sheet"?*

Right. We have followed your suggestion, thanks.

*P. 12, l. 31: "ISMIP6-Greenland _accounts_ for the vertical"*

Changed.

*P. 12, l. 34: "under _the_ RCP8.5 scenario"*

Added.

*P. 13, l. 1: "smaller compared to"*

Corrected.

*P. 13, l. 2: "respectively)_,_"*

Added.

*P. 13, l. 3: "smaller _than that of_ Vizcaino"*

Corrected.

*P. 20, Fig. 2, caption: "_with respect to_ the end"*

Corrected.

*P. 21, Fig. 3, caption: "for _the_ velocity difference_s_"*

Corrected.

---

## Author Response (AR3)

Dear Dr. Ayako Abe-Ouchi,

thank you very much for your time as Editor on our paper. We have performed a careful reading to correct the remaining typographical errors. Please find in this response a track-change version of the manuscript.

Best regards,
Aurélien Quiquet and Christophe Dumas

[revised manuscript text omitted]